# GLEAN: Guideline-Grounded Evidence Accumulation for High-Stakes Agent Verification

**Yichi Zhang** [1 2]  **Nabeel Seedat** [2 3]  **Yinpeng Dong** [1 4]  **Peng Cui** [1]  **Jun Zhu** [1]  **Mihaela van der Schaar** [2]

## Abstract

As LLM-powered agents have been used for high-stakes decision-making, such as clinical diagnosis, it becomes critical to develop reliable verification of their decisions to facilitate trustworthy deployment. Yet, existing verifiers usually underperform owing to a lack of domain knowledge and limited calibration. To address this, we establish **GLEAN**, an agent verification framework with **G**uide**L**ine-grounded **E**vidence **A**ccumulatio**N** that compiles expert-curated protocols into trajectory-informed, well-calibrated correctness signals. GLEAN evaluates the step-wise alignment with domain guidelines and aggregates multi-guideline ratings into surrogate features, which are accumulated along the trajectory and calibrated into correctness probabilities using Bayesian logistic regression. Moreover, the estimated uncertainty triggers active verification, which selectively collects additional evidence for uncertain cases via expanding guideline coverage and performing differential checks. We empirically validate GLEAN with agentic clinical diagnosis across three diseases from the MIMIC-IV dataset, surpassing the best baseline by 12% in AUROC and 50% in Brier score reduction, which confirms the effectiveness in both discrimination and calibration. In addition, the expert study with clinicians recognizes GLEAN's utility in practice.

*Figure 1.* **Guideline-grounded verification in clinical diagnosis.** For an agent clinician with access to different examinations (bottom), GLEAN verifies its diagnosis by assessing alignment with clinical guidelines (top) at each step. The example on a patient with acute diverticulitis illustrates how GLEAN accumulates guideline-grounded evidence into calibrated correctness probabilities along the execution (middle). While initial history at the first step aligns well with criteria, physical examination contradicts guidelines on abdominal tenderness and fever, dropping confidence to 0.5. Then, laboratory results recover the confidence, and CT imaging at the last step further confirms the diagnosis with higher confidence.

## 1. Introduction

Large Language Models (LLMs) are increasingly serving as autonomous agents (Yao et al., 2023; Park et al., 2023; Liu et al., 2024) in high-stakes domains such as healthcare (Tang et al., 2024; Fan et al., 2025). As they are entrusted in these

[1]Tsinghua University [2]University of Cambridge [3]Thomson Reuters Foundational Research [4]Shanghai Qi Zhi Institute. Correspondence to: Yinpeng Dong <dongyinpeng@tsinghua.edu.cn>, Jun Zhu <dcszj@tsinghua.edu.cn>, Mihaela van der Schaar <mv472@cam.ac.uk>.

*Proceedings of the 43rd International Conference on Machine Learning*, Seoul, South Korea. PMLR 306, 2026. Copyright 2026 by the author(s).

open-ended, risk-sensitive tasks, erroneous answers could lead to severe consequences in the real world. Therefore, it becomes critical to establish reliable verification that judges the correctness of decisions made by agents (Venktesh et al., 2025). In clinical diagnosis, for example, mistakes in agent decisions should be flagged before they are acted upon in patient care (Graber, 2013; Davenport & Kalakota, 2019). Yet, high-stakes domains usually present a fundamental asymmetry that generation is cheap, while verification is intrinsically harder, as accurate checking often requires domain expertise (Arora et al., 2025). This raises a central question: *How can we compile domain knowledge into reliable verification signals with calibrated correctness probabilities*, enabling risk control via abstention or escalation?

While verification has been studied for tasks like fact checking (Lin et al., 2024) and mathematics (Weng et al., 2023), these approaches are still insufficient for high-stakes agents. Reward modeling is a popular paradigm that trains a verifier to score an agent's outputs (Lambert et al., 2025; Xia et al., 2025) or its execution steps (Lightman et al., 2024; Zheng et al., 2025; Xi et al., 2025). While it can internalize domain knowledge from large-scale training data, obtaining substantial expert-labeled annotations is often prohibitively expensive and practically challenging. Meanwhile, training-free alternatives, including model-based ratings like LLM-as-a-Judge (Zheng et al., 2023) and sampling-based estimates such as self-consistency (Wang et al., 2023), are still inadequate. These techniques are weakly informed by explicit knowledge, leading to evaluations that are either biased toward implicit criteria in the model (Gu et al., 2024) or easily misled by consistent errors (Hobelsberger et al., 2025; Tan et al., 2025). Some approaches incorporate external examinations (Pan et al., 2023; Jiang et al., 2024), but they rarely yield a well-calibrated signal grounded in domain standards.

In this paper, we underscore that much of the necessary domain knowledge in high-stakes scenarios is already codified in professional protocols. They come in diverse forms, such as clinical guidelines, checklists, and operating procedures (Grimshaw & Russell, 1993; Shah et al., 2023), which could help reduce reliance on exhaustive sample-specific annotations and mitigate bias from implicit preferences through domain grounding. As exemplified in Figure 1, these structured materials often specify how decisions should be made and what constitutes an acceptable process in principle, naturally providing explicit and auditable standards to verify agent decisions step by step along the execution trajectories, instead of only judging their final outcomes. This motivates us to build calibrated signals upon these domain guidelines for verification, providing trajectory-informed estimates of the probability that the agent's decision is correct.

To this end, we introduce **GuideLine-grounded Evidence AccumulatioN (GLEAN)**, a novel verification framework for high-stakes agentic decision-making that instantiates domain guidelines into calibrated verification signals for agent execution. Taking a probabilistic formulation of sequential evidence accumulation along the agent trajectories, GLEAN obtains scores indicating alignment with multiple guidelines for each step and accumulates them as surrogate evidence for trajectory correctness. These guideline-grounded signals are shown to be informative and demonstrate an approximate monotonic linearity with correctness in the logit space, which allows us to acquire well-calibrated correctness probabilities via Bayesian logistic regression in a lightweight manner. However, the surrogate evidence can still be insufficient when the guidelines are incomplete or less specific. To address this, GLEAN further supports active verification using estimated uncertainty to selectively increase verification

effort at inference, analogous to the view of test-time scaling (Zhang et al., 2025b). Specifically, GLEAN performs guideline expansion for broader coverage and differential checks against competitive alternatives on uncertain cases, to gather additional evidence for better verification.

We validate GLEAN with clinical diagnosis (Hager et al., 2024), a representative high-stakes scenario that is error-sensitive, open-ended, and governed by explicit guidelines. We summarize our contributions as: ① ***Conceptually***: We reframe high-stakes agent verification as sequential evidence accumulation grounded in domain knowledge, which yields informative process signals that trigger active evidence collection, linking verification to test-time scaling. ② ***Technically***: We operationalize guidelines into per-step alignment scores, transform them via Bayesian logistic regression into calibrated correctness probabilities, and introduce active verification that refines verification signals when uncertainty is high. ③ ***Empirically***: We demonstrate the effectiveness of GLEAN across three disease diagnosis tasks, showing that GLEAN significantly outperforms popular verification methods in both discrimination and calibration, achieving AUROC over 0.94 and Brier scores lower than 0.10 with active verification. It also boosts agent diagnosis accuracy from 55.6% to 77.5% with Best-of-N. We also conduct an expert study with clinicians to confirm its practical utility, supporting reliable deployment in high-stakes settings.

## 2. Related Work

In this section, we briefly review the relevant literature on LLM-powered agents and verification methods.

### 2.1. LLM-powered Agents

LLMs have been deployed as autonomous agents (Yao et al., 2023) for multi-step problem solving, spanning software engineering (Jimenez et al., 2024), web browsing (Nakano et al., 2021), and computer use (Sager et al., 2025). They are also being explored in high-stakes domains such as finance (Yang et al., 2024a; Zhang et al., 2024a) and healthcare (Jin et al., 2019; Singhal et al., 2025). Recent systems increasingly target medical decision-making (Tang et al., 2024; Kim et al., 2024; Chen et al., 2025a; He et al., 2025), with clinical diagnosis as a representative task (Fan et al., 2025; Kyung et al., 2025). However, evaluations show a substantial gap compared to human experts (Hager et al., 2024; Chiu et al., 2025), and their errors are complicated by the multi-step nature, arising from intermediate steps (Zhu et al., 2025; Zhang et al., 2025c). Therefore, we study verification for agentic decision-making, with clinical diagnosis as a representative high-stakes application, where reliable estimates of correctness probabilities are required for trustworthy deployment (Banerji et al., 2023).

## 2.2. LLM and Agent Verification

Verification methods assess the reliability of model outputs and agent decisions. In high-stakes settings, verifiers should both discriminate correctness with domain expertise and provide well-calibrated estimates for risk-aware use. Prior work mainly focuses on verifying LLM answers, especially for fact-checking (Lin et al., 2024) and mathematics (Weng et al., 2023). Learned verifiers, such as outcome and process reward models (Lightman et al., 2024; Pandit et al., 2025; Thatikonda et al., 2025), can internalize expert preferences at scale, but require costly annotations and may not generalize under domain shift (Yang et al., 2024b). Training-free approaches are simpler but still limited. Model-based rating, e.g., token probability (Kadavath et al., 2022) and generative score (Zheng et al., 2023), depends on the internal knowledge and is often biased or weakly calibrated (Gu et al., 2024; Tian et al., 2025). Sampling-based signals from self-consistency (Wang et al., 2023; Manakul et al., 2023) and semantic entropy (Kuhn et al., 2023; Farquhar et al., 2024) reflect variability in sampled outputs, but can be overconfident under consistent mistakes (Hobelsberger et al., 2025; Tan et al., 2025). External examinations, including retrieval-based checks (Pan et al., 2023; Zhang et al., 2024b) and logical verification (Ling et al., 2023; Jiang et al., 2024), can provide additional evidence, but often yield heterogeneous cues rather than calibrated confidence.

In agentic workflows, beyond training reward models (Wang et al., 2025; Yun et al., 2025; Jiang et al., 2025), verification is often implemented with agent verifiers (Lifshitz et al., 2025; Zhuge et al., 2025) and environment-specific checkers. Examples include executing tests for code agents (Huang et al., 2023), validating actions in computer-use trajectories (Lu et al., 2025), and enforcing security or access-control policies as safety guardrails (Xiang et al., 2025; Chen et al., 2025b). While effective in certain tasks, these mechanisms do not directly address open-ended, high-stakes decision-making, where domain expertise must be explicit and calibrated confidence is needed for risk management. In contrast, we provide a framework that grounds verification in domain guidelines, producing trajectory-informed calibrated confidence for these scenarios.

# 3. Guideline-Grounded Evidence Accumulation

In this section, we present GLEAN as a guideline-grounded verification framework for open-ended, high-stakes agentic decision-making, which operationalizes domain protocols into well-calibrated signals of correctness and facilitates active verification under high uncertainty.

## 3.1. Verification as Sequential Evidence Accumulation

In agentic workflows, decisions and answers are essentially derived from the execution trajectories involving rationales and actions. Motivated by this multi-step nature in agent execution, we formulate agent verification as sequential evidence accumulation, where each step contributes incremental information toward the correctness of the final output.

Given an initial input, an agent interacts with the environment for $T$ steps, producing a trajectory with observations and actions $\tau_{1:T} = \{o_t, a_t\}_{t=1}^T$ and a final output $y$. We introduce a binary latent variable $Z \in \{1, 0\}$ indicating whether $y$ is correct or not. At each step $t$, we maintain a posterior probability of correctness conditioned on the trajectory prefix and the final prediction, defined as

$$p_t := P(Z = 1 | \tau_{1:t}, y), \tag{1}$$

where $\tau_{1:t}$ is the $t$-step prefix. This quantifies the verifier's confidence that the ongoing trajectory leading to the final answer is right. In our work, we ultimately use $p_T$ as the signal for such confidence in correctness to verify the final output. Below, we omit $y$ in condition for notation simplicity.

Using Bayes' rule, we derive a form of sequential evidence accumulation to decompose the correctness probability into step-wise information. We take the logit of the posterior to avoid calculating the partition function (Gold & Shadlen, 2007; Karamched et al., 2020), following

$$\underbrace{\log \frac{p_t}{1 - p_t}}_{\ell_t} = \underbrace{\log \frac{p_{t-1}}{1 - p_{t-1}}}_{\ell_{t-1}} + \underbrace{\log \frac{P(o_t, a_t \mid Z = 1, \tau_{1:t-1})}{P(o_t, a_t \mid Z = 0, \tau_{1:t-1})}}_{e_t},$$

$$\tag{2}$$

where $e_t$ represents the incremental evidence at step $t$ and can be gathered through this additive structure to an accumulated evidence $\ell_t$ for verification. Importantly, this formulation is inherently probabilistic. If $e_t$ and $\ell_t$ were available, the final estimate $p_T$ would be well-calibrated.

## 3.2. Guideline-Grounded Surrogate Evidence

However, in open-ended agentic settings, $e_t$, computed with the likelihood of observations and actions, is generally intractable, since these variables can be high-dimensional and partially described. Intuitively, $e_t$ captures how the current step supports an acceptable outcome, which motivates us to construct step-wise surrogate signals that assess whether the agent's behavior is reasonable for the answer and align them with well-calibrated confidence for correctness.

This corresponds to our motivation in Section 1 that, in high-stakes professional scenarios, guidelines exist as explicit standards that practitioners follow to make decisions. They are usually expert-curated protocols and auditable criteria, specifying what constitutes proper evidence and providing

clear instructions on actions under the domain rules, which make them a potential source for step-wise verification signals grounded in external knowledge.

We acquire such signals with an implementation of model-based rating. For each trajectory, we retrieve a guideline $g$ relevant to the given context and the final answer from an external guideline set $\mathcal{G}$. At each step $t$ with $(o_t, a_t)$ after a prefix $\tau_{1:t-1}$, we prompt an LLM judge $J$ to score the current step given the corresponding guideline. Specifically, following a common practice (Kadavath et al., 2022; Fanconi & van der Schaar, 2025), we ask the judge whether the step following the history aligns with the provided guideline and extract the token probability over a discrete label set $\{\texttt{YES}, \texttt{NO}\}$, resulting in a scalar rating $s_{t,g} =$

$$\frac{\mathcal{J}(\texttt{YES}|\tau_{1:t-1}, o_t, a_t, g)}{\mathcal{J}(\texttt{YES}|\tau_{1:t-1}, o_t, a_t, g) + \mathcal{J}(\texttt{NO}|\tau_{1:t-1}, o_t, a_t, g)}. \quad (3)$$

As model-based ratings tend to be miscalibrated and lack reliable probabilistic semantics, we need to learn a calibration function that maps them to signals reflecting well-calibrated probabilities. Since ground-truth $e_t$ is generally unavailable, we do not have step-wise supervision, but instead, only have a trajectory-level correctness label $Z$ for the final outcome, which defines the accumulated evidence $\ell_t$ in Equation (2). Accordingly, we construct a prefix-level surrogate evidence $S_t$ from the step-wise ratings $\{s_{i,g}\}_{i=1}^{t}$ and learn to calibrate it to align $\ell_t$. This inherently encourages $S_t$ to aggregate step-wise contributions to the accumulated evidence.

In practice, supervision remains limited in high-stakes scenarios, even for trajectory-level correctness labels, making simple and data-efficient calibration essential. We hereby discuss the properties required of $S_t$ for lightweight calibration, and then show that guideline grounding indeed provides more information and a better potential for this.

**Remark.** For any prefix signal $S_t$ in a trajectory, if it meets

1. *Approximate Sufficiency:* $S_t$ captures most of the information in the prefix for predicting correctness, i.e., $P(Z = 1 \mid \tau_{1:t}) \approx P(Z = 1 \mid S_t)$;
2. *Monotonic near-linearity:* $S_t$ exhibits a monotonically increasing and near-linear relationship with correctness in logit space, i.e., $\log \frac{P(Z=1|S_t)}{1-P(Z=1|S_t)} \approx aS_t + c, a > 0$,

then a low-capacity linear calibrator suffices to provide an adequate estimate of $\ell_t$ from $S_t$ with scarce supervision. We present an analysis of the error bound in Appendix A.1.

**Do guideline-grounded signals satisfy these?** Motivated by the additivity of evidence in logit space and the common practice of logit-based calibration (Guo et al., 2017; Kull et al., 2017), we study a simple construction for the accumulated surrogate evidence $S_t = \sum_{i=1}^{t} \log \frac{s_{i,g}}{1-s_{i,g}}$, which sums up the logits of step-wise guideline-grounded ratings for a prefix $\tau_{1:t}$. Empirically, we validate the desired prop-

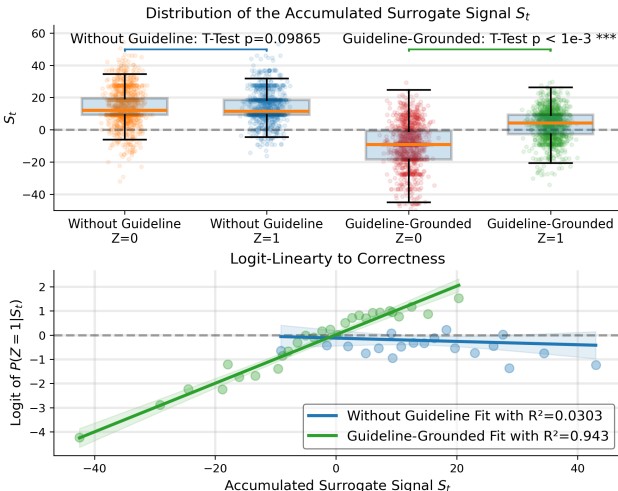

*Figure 2.* **Properties of guideline-grounded signals.** Top: Signals grounded in guidelines significantly discriminate correct from incorrect prefixes, while uninformed signals do not. Bottom: Guideline-grounded signals exhibit strong logit-linearity with correctness, but uninformed ones lack this property.

erties of this signal $S_t$ in agentic clinical diagnosis (Hager et al., 2024). Concretely, we sample 5,000 prefixes from trajectories generated by Qwen3-30B-A3B-Instruct (Yang et al., 2025) and examine the guideline-grounded surrogate evidence, where the judge is provided with a guideline $g = g_y \in \mathcal{G}$ retrieved based on the final diagnosis $y$. Besides, we also include an uninformed variant, where the judge provides ratings based on its internal knowledge, i.e., $g = \varnothing$, to better examine the contribution of guidelines.

Subsequently, we examine whether the accumulated surrogate evidence $S_t$ is discriminative of correctness and whether the logit of $P(Z = 1|S_t)$ is linearly related to $S_t$. As shown in Figure 2, signals without guidelines do not yield statistically significant separation between correct and incorrect prefixes, and they demonstrate a weak monotonic or logit-linear trend with correctness. This suggests that model-based ratings alone can be unreliable as verification signals, confirming the issue of biased and inconsistent evaluation (Gu et al., 2024). In contrast, guideline grounding substantially enhances both desired properties, showing highly significant discrimination between prefixes ($p < 1e-3$) and an approximately linear relationship with correctness in logit space ($R^2 = 0.943$). Together, these results support guideline-grounded signals as informative surrogate evidence, which also confirms our motivation of using guidelines as external knowledge for verification, and further justify the employment of a simple linear calibrator to obtain well-calibrated probabilistic estimates.

### 3.3. Robust and Active Evidence Accumulation

While Section 3.2 suggests that guideline-grounded scores can be informative and approximately logit-linear, the suffi-

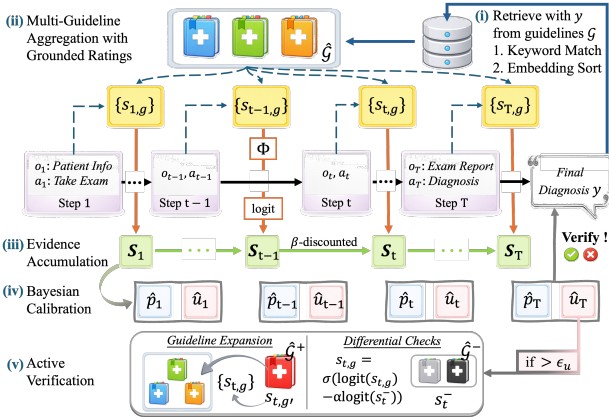

*Figure 3.* **Pipeline of GLEAN for clinical diagnosis.** GLEAN (i) retrieves guidelines for the final diagnosis, (ii) aggregates step-wise scores for alignment with multiple guidelines, (iii) accumulates them into $\beta$-discounted evidence, which is (iv) calibrated to yield confidence and uncertainty. (v) High uncertainty then triggers active verification via guideline expansion and differential checks.

ciency of a single rating from only one guideline can vary in practice due to noisy judgments and imperfect guideline retrieval, which motivates us to develop more reliable surrogate evidence, composing our practical framework of GLEAN as summarized in Figure 3 and Algorithm 1.

**Multi-Guideline Aggregation.** Beyond single-guideline scoring, we use a set of retrieved guidelines and aggregate their step-wise ratings into robust statistics to reduce variance in grounded evidence. For each trajectory, we can acquire a group of guidelines $\hat{\mathcal{G}} \in \mathcal{G}$ relevant to support verification and get multiple ratings $\{s_{t,g}\}_{g\in\hat{\mathcal{G}}}$ accordingly for every step. To preclude the impact from the varying numbers of available guidelines, we aggregate the guideline ratings into a single step-level feature

$$\mathbf{s}_t = \Phi(\{s_{t,g}\}_{g\in\hat{\mathcal{G}}}) \in [0,1]^d, \qquad (4)$$

where $d$ statistics (e.g., average, minimum) are extracted from the ratings. Given the availability of guidelines and the verification budget, we can set the number of guidelines for rating with $|\hat{\mathcal{G}}| = K$. Note that verification with one single guideline is a special case of our method, where the rating is a scalar feature. We then accumulate the step-level features into a prefix-level signal that summarizes the grounded evidence up to the current step. To better mitigate noises introduced by early-step deviations due to limited information, we apply a discounted sum for evidence accumulation in practice, i.e., $\mathbf{S}_t = \sum_{i=1}^{t} \beta^{t-i} \log \frac{\mathbf{s}_i}{1-\mathbf{s}_i}$, with $\beta$ as the discount factor. We show that the discounted accumulation does not alter the properties in Appendix A.2.

As indicated in Section 3.2, while the scale of $\mathbf{S}_t$ is not in $[0,1]$, it can be mapped to the probability of correctness through calibration with a linear model. Given a labeled

dataset $\mathcal{D}$ with evidence $\mathbf{S}_t$ and label $Z$ as a sample, we adopt Bayesian logistic regression for the calibrator:

$$P(Z = 1 \mid \mathbf{S}_t, \mathbf{w}, b) = \sigma(\mathbf{w}^\top \mathbf{S}_t + b), \qquad (5)$$

where $\sigma$ denotes the sigmoid function. We set 0-centered Gaussian priors, $\mathbf{w} \sim \mathcal{N}(\mathbf{0}, \lambda^{-1}\mathbf{I}_d)$ and $b \sim \mathcal{N}(0, \lambda^{-1})$, for the parameters $\mathbf{w}$ and $b$, which is equivalent to standard logistic regression with an $\ell_2$ regularization after taking expectations over the posterior. We draw samples from their posterior $p(\mathbf{w}, b|\mathcal{D})$ with Markov chain Monte Carlo (MCMC) (Andrieu et al., 2003) and obtain a calibrated probability estimate based on the expectation over this distribution. Concretely, we provide a confidence for verification following

$$\hat{p}_T = \mathbb{E}_{\mathbf{w},b\sim p(\cdot|\mathcal{D})}[\sigma(\mathbf{w}^\top \mathbf{S}_T + b)], \qquad (6)$$

where $S_T$ is the accumulated signal up to the final answer.

**Uncertainty-Triggered Active Verification.** While aggregating multiple guidelines improves robustness to randomness and single-guideline bias, surrogate evidence can still be insufficient for difficult cases, especially when subtle errors are missed due to incomplete coverage or when the evidence is not sample-specific enough. Fortunately, well-calibrated probabilities yield reliable uncertainty estimates, which allow us to trigger active verification appropriately. If the uncertainty $\hat{u}_T$ at the last step (e.g., entropy $H(\hat{p}_T)$) exceeds a threshold $\epsilon_u$, we can collect additional evidence to strengthen verification. Below, we introduce two complementary strategies targeting the aforementioned limitations.

To address the insufficient information provided by the collected guidelines, we conduct *guideline expansion*. We broaden the evidence pool by retrieving additional guidelines that are relevant to the current context or the final answer. Let $\hat{\mathcal{G}}^+$ denote the expanded guideline set. When we obtain ratings for the newly collected guidelines, we can correct the statistics over $\hat{\mathcal{G}} \cup \hat{\mathcal{G}}^+$ and the accumulated signals with the supplementary information from the new guidelines, leading to a re-evaluated confidence in the already learned calibrator. This strategy stabilizes the surrogate evidence by incorporating more knowledge for samples with uncertainty stemming from insufficient examination.

Meanwhile, to overcome the issue of weak specificity in retrieved guidelines, we perform *differential checks* with guidelines of competitive alternatives. Ideally, a trajectory with the final answer $y$ should align substantially better with guidelines that support $y$ than with those supporting competing outcomes. Otherwise, high scores may be driven by generally plausible standards that also fit alternatives. To rectify such false support, we actively retrieve competitive outcomes $\{y' \neq y\}$ and their corresponding guidelines $\hat{\mathcal{G}}^-$. For each step, we obtain competitive scores $\{s_{t,g}^-\}_{g\in\hat{\mathcal{G}}^-}$ and

correct each existing score in logit space with the maximum score $s_t^-$ across the competitive guidelines, following

$$\forall g \in \hat{\mathcal{G}}, s_{t,g}^{rec} = \sigma \left( \log \frac{s_{t,g}}{1 - s_{t,g}} - \alpha \log \frac{s_t^-}{1 - s_t^-} \right), \quad (7)$$

where $\alpha \geq 0$ is a rectification factor. The rectified scores are then used for subsequent aggregation and evidence accumulation. Intuitively, strong competitive alignment decreases the effective support for the answer, incorporating counter-evidence and preventing over-confidence when the trajectory is better explained by an alternative outcome. Taking medical diagnosis as an example, there are often potential alternative diagnoses, i.e., differential diagnoses, that partially share similar symptoms but imply different treatments. We extract these competitive predictions from the candidate rollouts and retrieve the corresponding guidelines as counter-evidence.

## 4. Experiments

In this section, we evaluate our verification framework in agentic clinical diagnosis (Hager et al., 2024), a representative high-stakes decision-making setting, to show its effectiveness. Diagnosis is typically process-informed and risk-sensitive, and it is rich in protocols, with publicly available clinical guidelines serving as explicit, auditable standards for verification, making it a natural testbed for GLEAN.

### 4.1. Experimental Setup

We hereby introduce experimental setups, covering datasets, implementation details, evaluation metrics, and baselines.

**Task and Datasets.** We study three MIMIC diseases (Johnson et al., 2023), i.e., diverticulitis, cholecystitis, and pancreatitis, using the ReAct-style agent workflow by Hager et al. (2024), where the agent iteratively reasons over patient information, requests examinations, and outputs a final diagnosis. Due to MIMIC restrictions on commercial LLMs, we use Qwen2.5-7B-Instruct and Qwen3-30B-A3B-Instruct as backbones, generating multiple trajectories per case at a temperature of 0.9. A trajectory is correct if its final diagnosis matches the ground-truth disease. For each backbone, we evaluate 2000 trajectories with a 50/50 correct–incorrect split, balanced across cases and diseases. We use the guideline dataset[1] from medical pretraining (Chen et al., 2023), which contains high-quality clinical practice guidelines.

**Implementation of GLEAN.** For controlled evaluation, we use the same backbone as the agent for GLEAN and methods with model-based rating. We provide the detailed prompt for step-wise rating in Appendix B.1. We retrieve guidelines by keyword-matching with titles and then rerank them by `all-mpnet-base-v2` embedding similarity.

We fix the number of guidelines used for aggregation to $K = 1, 3$, with aggregation $\Phi(\{s_{t,g}\}) = [\min, \operatorname{avg}]_{g \in \hat{\mathcal{G}}} s_{t,g}$. The Bayesian logistic regression calibrator is trained on 100 labeled trajectories separate from the evaluation set and we draw 2000 times from the posterior to obtain calibrated confidence and uncertainty. We use entropy for uncertainty and perform active verification using the calibrator with $K = 3$. For triggered samples, we add one guideline for expansion and two competitive guidelines for differential checks. We set factors $\beta = 0.5$ and $\alpha = 0.2$.

**Metrics.** To evaluate verification signals by GLEAN, we report AUROC for discrimination, Risk@0.5 as the error rate on the top 50% most confident samples, and calibration metrics, including Expected calibration error (ECE) (Naeini et al., 2015) and the Brier score (Glenn et al., 1950). We also assess GLEAN's utility via Best-of-N. For each case, we sample 16 trajectories and select the one scored the highest by the verifier, reporting accuracy for N∈ $\{4, 8, 16\}$.

**Baselines.** We compare GLEAN against diverse baselines. We take $P(\text{TRUE})$ (Kadavath et al., 2022) that uses the token likelihood of correctness from the verifier, and LLM-as-a-Judge (Zheng et al., 2023) with prompted generative scoring for model-based rating methods. Sampling-based baselines include Self-Consistency (Wang et al., 2023) and Semantic Entropy (Farquhar et al., 2024), which derive confidence from agreement and distributional entropy over 16 sampled trajectories. Besides, we adopt Self-Verification (Weng et al., 2023), which explicitly re-checks whether the answer supports the observations, and RAG-augmented rating, which incorporates the same retrieved guideline based on $P(\text{TRUE})$, as typical methods for external examinations. For learned verifiers, due to the lack of reward models for agentic clinical diagnosis, we take Med-PRM (Yun et al., 2025) as an example, which was trained on medical QA and also train an outcome reward model (ORM) with Qwen2.5-7B-Instruct on the same calibration set as GLEAN.

### 4.2. Main Results

Table 1 summarizes verification performance across three diseases and two agent backbones. Overall, GLEAN shows competitive performance with a single retrieved guideline per case ($K = 1$), yielding stronger discrimination (AUROC>0.82) and improved calibration (ECE<0.11). In contrast, baselines that mainly rely on the verifier's implicit knowledge (e.g., $P(\text{True})$, LLM-as-a-Judge, and Self-Verification) underperform, indicating that internal criteria alone are insufficient for reliable high-stakes diagnosis verification. Sampling-based verifiers are notably less effective on harder diseases such as cholecystitis and pancreatitis, where consistent yet incorrect generations degrade discrimination. RAG-augmented verification confirms the benefits of guideline retrieval but still lags behind GLEAN, high-

---

[1] https://huggingface.co/datasets/epfl-llm/guidelines

*Table 1.* **Verification performance across three diseases for two agent backbones.** We report AUROC, Risk@0.5, ECE, and Brier. We omit calibration metrics for Semantic Entropy since it is not a probability. For GLEAN, we report active verification with an entropy threshold of 0.5, and also a zero-threshold setting as an upper bound (gray). Best results are in **bold**, and second-best are underlined.

| Method | Diverticulitis | | | | Cholecystitis | | | | Pancreatitis | | | |
|---|---|---|---|---|---|---|---|---|---|---|---|---|
| | AUROC ↑ | Risk@0.5 ↓ | ECE ↓ | Brier ↓ | AUROC ↑ | Risk@0.5 ↓ | ECE ↓ | Brier ↓ | AUROC ↑ | Risk@0.5 ↓ | ECE ↓ | Brier ↓ |
| Agent Backbone: *Qwen2.5-7B-Instruct* | | | | | | | | | | | | |
| $P(\textsc{True})$ | 0.7280 | 0.3642 | 0.3445 | 0.3379 | 0.6225 | 0.2810 | 0.3640 | 0.3715 | 0.6565 | 0.5610 | 0.4009 | 0.4103 |
| LLM-as-a-Judge | 0.7709 | 0.3457 | 0.2558 | 0.2859 | 0.6426 | 0.2557 | 0.0917 | 0.2259 | 0.6528 | 0.5523 | 0.3701 | 0.3549 |
| Self-Consistency | 0.9011 | 0.2160 | 0.1880 | 0.1677 | 0.7978 | 0.1646 | 0.0834 | 0.1484 | 0.7621 | 0.4709 | 0.2582 | 0.2446 |
| Semantic Entropy | 0.8573 | 0.2531 | – | – | 0.7772 | 0.1747 | – | – | 0.7174 | 0.4884 | – | – |
| Self-Verification | 0.7891 | 0.3148 | 0.1198 | 0.2012 | 0.7576 | 0.1722 | 0.2210 | 0.2299 | 0.6809 | 0.5262 | 0.1071 | 0.2298 |
| RAG-Augmented | 0.8358 | 0.2284 | 0.2304 | 0.2271 | 0.7571 | 0.1899 | 0.2491 | 0.2598 | 0.8461 | 0.4186 | 0.2484 | 0.2531 |
| Med-PRM | 0.7111 | 0.3704 | 0.4851 | 0.4703 | 0.6475 | 0.2582 | 0.3010 | 0.3107 | 0.5877 | 0.5930 | 0.5797 | 0.5608 |
| ORM | 0.7793 | 0.3203 | 0.1594 | 0.2168 | 0.8190 | 0.1364 | 0.1283 | 0.1846 | 0.7002 | 0.5108 | 0.1964 | 0.2434 |
| GLEAN ($K=1$) | 0.8919 | 0.2469 | **0.0502** | 0.1359 | 0.8524 | 0.1089 | **0.0641** | 0.1495 | 0.8248 | 0.4331 | 0.0933 | 0.1714 |
| GLEAN ($K=3$) | 0.9568 | 0.1667 | 0.0724 | 0.0892 | 0.8929 | 0.0987 | 0.0893 | 0.1174 | 0.9078 | 0.3582 | 0.0994 | 0.1318 |
| +Active ($\epsilon_u = 0.5$) | **0.9756** | **0.1049** | 0.0686 | **0.0586** | **0.9063** | **0.0962** | 0.1067 | **0.1124** | **0.9325** | **0.3343** | **0.0701** | **0.1031** |
| +Active ($\epsilon_u = 0.0$) | 0.9869 | 0.1049 | 0.0673 | 0.0522 | 0.9159 | 0.0886 | 0.1206 | 0.1116 | 0.9471 | 0.3211 | 0.0760 | 0.0953 |
| Agent Backbone: *Qwen3-30B-A3B-Instruct* | | | | | | | | | | | | |
| $P(\textsc{True})$ | 0.7851 | 0.2840 | 0.2392 | 0.2488 | 0.7384 | 0.2222 | 0.2758 | 0.2927 | 0.7510 | 0.4519 | 0.2738 | 0.2815 |
| LLM-as-a-Judge | 0.8166 | 0.2654 | 0.1172 | 0.1910 | 0.7548 | 0.2197 | **0.0657** | 0.2007 | 0.7663 | 0.4169 | 0.1488 | 0.2083 |
| Self-Consistency | 0.9131 | 0.1667 | 0.1889 | 0.1608 | 0.8493 | 0.1439 | 0.1019 | 0.1534 | 0.8232 | 0.3732 | 0.2088 | 0.2096 |
| Semantic Entropy | 0.8128 | 0.2346 | – | – | 0.8448 | 0.1566 | – | – | 0.7991 | 0.3644 | – | – |
| Self-Verification | 0.7681 | 0.3333 | 0.2416 | 0.2655 | 0.7907 | 0.2045 | 0.1495 | 0.2057 | 0.7567 | 0.4344 | 0.3419 | 0.3241 |
| RAG-Augmented | 0.8577 | 0.2222 | 0.1816 | 0.1916 | 0.7896 | 0.1894 | 0.2575 | 0.2709 | 0.8590 | 0.3353 | 0.1671 | 0.1841 |
| Med-PRM | 0.8099 | 0.2531 | 0.4815 | 0.4684 | 0.7519 | 0.2146 | 0.3670 | 0.3626 | 0.6113 | 0.5277 | 0.5654 | 0.5515 |
| ORM | 0.8228 | 0.2614 | 0.1939 | 0.2381 | 0.8332 | 0.1555 | 0.1922 | 0.2146 | 0.7146 | 0.4492 | 0.2109 | 0.2670 |
| GLEAN ($K=1$) | 0.9147 | 0.1790 | 0.1077 | 0.1263 | 0.8888 | 0.1111 | 0.0849 | 0.1314 | 0.8367 | 0.3586 | 0.0953 | 0.1696 |
| GLEAN ($K=3$) | 0.9794 | 0.0741 | 0.1051 | 0.0632 | 0.9559 | 0.0505 | 0.0718 | 0.0790 | 0.8899 | 0.3003 | 0.0777 | 0.1350 |
| +Active ($\epsilon_u = 0.5$) | **0.9862** | **0.0370** | 0.1031 | **0.0453** | **0.9582** | **0.0480** | 0.0696 | **0.0753** | **0.9061** | **0.2828** | 0.0747 | **0.1194** |
| +Active ($\epsilon_u = 0.0$) | 0.9975 | 0.0309 | 0.1110 | 0.0393 | 0.9648 | 0.0379 | 0.0750 | 0.0753 | 0.9119 | 0.2915 | 0.0722 | 0.1175 |

lighting the importance of trajectory-aware evidence accumulation and explicit calibration. Finally, Med-PRM shows limited transfer to this agentic diagnosis setting, likely due to task and domain mismatch, while ORM achieves better results due to additional training, which are still inferior to GLEAN with the same amount of training data. This confirms the data efficiency provided by guideline grounding.

Building on the strong results with a single guideline, multi-guideline aggregation ($K=3$) further improves robustness, consistently increasing AUROC and reducing Risk@0.5 as well as Brier score, without significantly changing ECE. For instance, on Qwen3-30B for Diverticulitis, aggregation boosts AUROC from 0.9147 to 0.9789 while reducing Risk@0.5 from 0.1790 to 0.0802, with Brier dropping from 0.1263 to 0.0647. With a threshold of $\epsilon_u = 0.5$, active verification further improves AUROC and reduces risk beyond passive aggregation, with AUROC reaching 0.9856 and Risk@0.5 dropping to 0.0494 for the former example, while it also helps with calibration in terms of the Brier score, suggesting that the selectively collected evidence better facilitates verification under high uncertainty. We also

provide results with $\epsilon_u = 0.0$, which suggests a performance upper bound when active verification is applied to all samples, indicating the potential with sufficient budget.

### 4.3. Detailed Analysis

We present analyses to confirm the effectiveness of our framework and justify the soundness of the design. Below, we report metrics by gathering samples from all diseases.

**Active Verification.** The ablation in Table 2 confirms that the improvements from active verification come from two complementary sources of evidence. While each strategy alone provides consistent improvements over multi-guideline aggregation, combining them yields the strongest overall performance, indicating that coverage and specificity errors are both salient for diagnosis. Further, Figure 5 shows curves of performance to trigger ratios. This verifies that increasing the number of samples improves AUROC and reduces risk, but the gains are mostly achieved with a relatively small trigger ratio. Beyond that, improvements become marginal as it approaches full activation. Together, these results support uncertainty-triggered active verifica-

*Table 2.* **Ablation on active verification strategies.**

| Strategy | AUROC ↑ | Risk@0.5 ↓ | ECE ↓ | Brier ↓ |
|---|---|---|---|---|
| Agent Backbone: *Qwen2.5-7B-Instruct* | | | | |
| GLEAN ($K = 3$) | 0.9137 | 0.1678 | **0.0418** | 0.1178 |
| w. GE | 0.9322 | 0.1422 | 0.0564 | 0.1050 |
| w. DC | 0.9304 | 0.1444 | 0.0565 | 0.1062 |
| w. Both | **0.9470** | **0.1300** | 0.0700 | **0.0947** |
| Agent Backbone: *Qwen3-30B-A3B-Instruct* | | | | |
| GLEAN ($K = 3$) | 0.9389 | 0.1267 | **0.0550** | 0.0975 |
| w. GE | 0.9467 | 0.1167 | 0.0625 | 0.0908 |
| w. DC | 0.9477 | 0.1144 | 0.0841 | 0.0995 |
| w. Both | **0.9557** | **0.1078** | 0.0653 | **0.0849** |

*Table 3.* **Ablations on component design and guideline quality.**

| Method | AUROC ↑ | Risk@0.5 ↓ | ECE ↓ | Brier ↓ |
|---|---|---|---|---|
| *Components in GLEAN* | | | | |
| $P(\texttt{TRUE})$ | 0.6497 | 0.3933 | 0.3719 | 0.3803 |
| + Process | 0.6935 | 0.3500 | 0.3655 | 0.3721 |
| + Guideline | 0.7944 | 0.2755 | 0.2613 | 0.2703 |
| + Accumulation | 0.8547 | **0.2367** | 0.2233 | 0.2367 |
| + Calibration | **0.8574** | 0.2378 | **0.0360** | **0.1554** |
| *Guideline Quality* | | | | |
| Empty Guideline | 0.6321 | 0.4000 | 0.0918 | 0.2416 |
| Short Abstract | 0.7573 | 0.3122 | 0.0626 | 0.2043 |
| Detailed Content | **0.8574** | **0.2378** | **0.0360** | **0.1554** |
| *Retrieval Quality* | | | | |
| 0% Corrupted | 0.8197 | 0.2722 | 0.0539 | 0.1732 |
| 10% Corrupted | 0.7938 | 0.2855 | 0.0313 | 0.1849 |
| 10% + Active | 0.8217 | 0.2467 | 0.0742 | 0.1774 |
| 30% Corrupted | 0.7647 | 0.3167 | 0.0327 | 0.1956 |
| 30% + Active | 0.8117 | 0.2500 | 0.0806 | 0.1845 |

tion as an effective form of test-time scaling.

**Best-of-N Selection.** Results in Figure 4 further highlight the practical value of verification signals from GLEAN as a ranking mechanism. While baselines often yield smaller and more saturating improvements, GLEAN, especially with multi-guideline aggregation, consistently improves performance, effectively exploiting additional sampled candidates. For Qwen3-30B, the accuracy increases steadily from 58.94% to 73.8% / 76.7% / 78.3%, while self-consistency only achieves 70.0%. This suggests that GLEAN is not only informative for correctness assessment, but also directly usable to improve practical agentic decision-making.

**Components in GLEAN.** The upper panel in Table 3 presents a component-wise ablation of our framework with Qwen2.5-7B by progressively adding key ingredients. Starting from the baseline of $P(\text{True})$ derived from the model's own token probability for the agent output, the trajectory context brings a modest improvement, suggesting that being process-informed helps verification to a limited extent. Incorporating guideline grounding leads to substantial improvements in discrimination and lower risk, highlighting the importance of introducing explicit domain knowledge rather than relying on implicit criteria. Further, accumulating step-wise evidence along the trajectory further strengthens performance. Eventually, after calibration, which comprises the complete GLEAN, while maintaining AUROC and Risk@0.5, it significantly improves ECE and Brier, mapping the surrogate evidence to well-calibrated estimates.

**Guideline Quality.** The middle panel in Table 3 studies the impact of guideline quality. Using only an empty placeholder performs poorly, while replacing it with a short abstract already yields a clear improvement across all metrics, indicating that even coarse protocol signals can aid verification. Importantly, detailed guideline content yields greater gains in both discrimination and calibration (e.g., AUROC from 0.7573 to 0.8574 and ECE from 0.0918 to 0.0360 for Qwen2.5-7B), demonstrating that richer and more specific domain protocols lead to more informative evidence.

**Retrieval Quality** It is critical to notice that imperfect re-

trieval is a realistic case in practice, and we propose active verification to explicitly address this by selectively expanding guideline coverage and introducing counter-evidence from competitive alternatives when the initial evidence is uncertain or insufficient. To directly study this, we conduct a controlled retrieval-corruption experiment, where a fraction of originally retrieved high-quality guidelines is replaced with less relevant ones, as displayed in the lower panel in Table 3. As retrieval quality degrades, GLEAN shows graceful degradation rather than collapsing abruptly. AUROC drops from 0.8197 to 0.7938/0.7647, and Risk@0.5 rises from 0.2722 to 0.2855/0.3167 under 10%/30% corruption. Importantly, active verification recovers a substantial portion of this loss in both settings, bringing AUROC back to 0.8270/0.8184 and reducing Risk@0.5 to 0.2455/0.2844. This supports our intended use of active verification as a practical mechanism for mitigating imperfect retrieval. At the same time, the calibration is not impacted by the imperfect retrieval quality, with insignificant changes for ECE and even slight improvements in Brier scores. Overall, active verification mainly improves discrimination and selective-risk performance, while having the calibrator stable under the corruption of guideline retrieval.

**Expert Study.** Three expert clinicians independently reviewed 50 agent trajectories with GLEAN's guideline-grounded confidence scores, annotating the first erroneous step and rating interpretability and clinical utility on 5-point scales. Overall usefulness of GLEAN as a useful verification tool was rated 4.67/5. Clinician's rated interpretability favorably ($M=4.36$, $SD=0.79$) and found confidence scores clinically useful ($M=4.12$, $SD=0.82$). Additionally, 87% of interpretability ratings and 90% of clinical utility ratings fell within one point, indicating substantive consensus between clinicians on GLEAN's value. Inter-rater reliability on diagnosis error identification was substantial (Cohen's

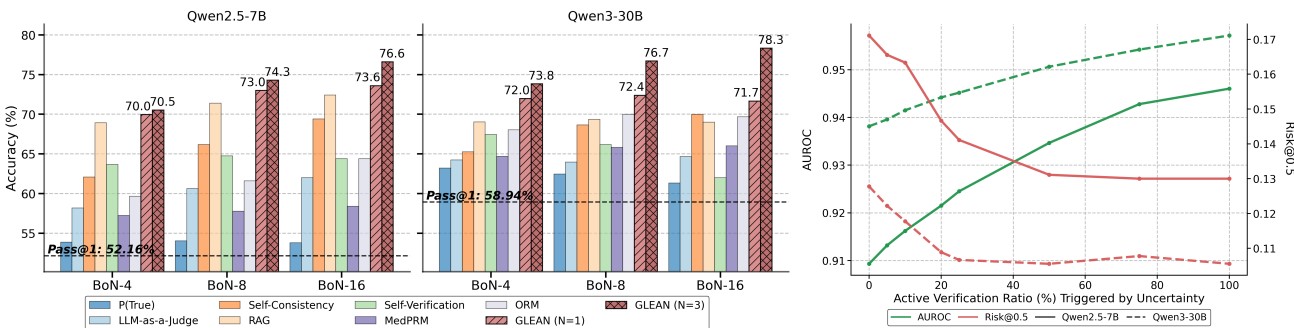

Figure 4. **Accuracy of Best-of-N selection using different signals.**

Figure 5. **Performance with active verification.**

$\kappa$=0.78). Clinicians also qualitatively noted that "confidence metrics were useful to know where to review" and that uncertainty "reflected the agent fixating on one possible diagnosis." Details of the expert study are in Appendix B.3

**Additional Experiments.** We provide additional experiments in Appendix B.2 on certain designs in GLEAN, cross-agent verification and computational costs, along with detailed results for Best-of-N selection.

### 4.4. Other Scenarios

To demonstrate the generalization of GLEAN across different tasks and scenarios, we adopt two more datasets, AgentClinic-MedQA (Schmidgall et al., 2024) and Agent-SafetyBench (Zhang et al., 2025a), which covers both different medical task and different agent verification objective.

Table 4. **Verification performance of GLEAN in other scenarios covering medical question answering and agent safety.** SC is short for self-consistency.

| Backbone | Method | AUROC | Risk@0.5 | ECE | Brier |
|---|---|---|---|---|---|
| *AgentClinic-MedQA* | | | | | |
| Qwen2.5-7B | P(True) | 0.7038 | 0.3000 | 0.3756 | 0.3706 |
| Qwen2.5-7B | SC | 0.6529 | 0.3000 | 0.2261 | 0.2760 |
| Qwen2.5-7B | GLEAN($N$=1) | 0.7606 | **0.2000** | 0.1666 | 0.2260 |
| Qwen2.5-7B | GLEAN($N$=3) | **0.7867** | 0.2200 | **0.0893** | **0.2080** |
| GPT-4.1 | P(True) | 0.7207 | 0.2460 | 0.1607 | 0.2167 |
| GPT-4.1 | SC | 0.7637 | 0.1600 | 0.2894 | 0.2724 |
| GPT-4.1 | GLEAN($N$=1) | 0.8121 | 0.1800 | 0.0733 | **0.2094** |
| GPT-4.1 | GLEAN($N$=3) | **0.8321** | **0.1200** | **0.0723** | 0.2338 |
| *Agent-SafetyBench* | | | | | |
| Qwen3-30B | P(True) | 0.6167 | 0.4360 | 0.3695 | 0.3818 |
| Qwen3-30B | LLM-Judge | 0.5458 | 0.4120 | 0.3851 | 0.3966 |
| Qwen3-30B | GLEAN | **0.7283** | **0.3556** | **0.0900** | **0.2137** |

We first apply GLEAN to AgentClinic-MedQA, where the agent is asked to answer medical questions by calling tools through an agentic manner. We show the results in the upper panel in Table 4. On Qwen2.5-7B, we observe similar trends as on MIMIC diseases. Grounded in medical guidelines, GLEAN achieves better verification accuracy and calibration, which is further improved when we retrieve

more guidelines to support the scoring. In addition to that, as MedQA is allowed to be run with commercial models, we also test GLEAN with GPT-4.1[2], an OpenAI-hosted LLM. The superiority of GLEAN also holds for this new model, indicating the generalization of GLEAN across tasks and models.

Moving beyond the domain of healthcare, we additionally evaluated GLEAN with Qwen3-30B on Agent-SafetyBench, a qualitatively different safety-critical agent setting, which is of great application concerns and with explicit guidelines on reliable deployment. As shown in the lower panel in Table 4, GLEAN brings a consistent improvement over baselines in terms of discrimination and calibration. This further confirms the contributions from guideline-grounded stepwise scoring along with the lightweight calibrator.

### 5. Conclusion

We introduce GLEAN, a guideline-grounded verification framework that achieves strong discrimination and calibration by operationalizing domain protocols into trajectory-informed confidence in the correctness of high-stakes agents, such as clinical diagnosis. It incorporates multiple strategies to enhance the informativeness of the verification signal, including multi-source aggregation and active evidence collection, demonstrating practical utility as confirmed by expert validation. GLEAN provides a formal method to include domain knowledge for verification, which is naturally potential to extend with codified standards or unstructured expert experience, such as legal, financial, or safety-critical settings, and provides a practically reliable solution for deploying autonomous agents across diverse high-stakes applications.

---

[2]https://openai.com/index/gpt-4-1/

## Acknowledgement

This work was supported by Fundamental and Interdisciplinary Disciplines Breakthrough Plan of the Ministry of Education of China (No. JYB2025XDXM101), NSF of China Projects (Nos. 62276149, 92370124, 92248303, U25B6003, 62550004, 92270001, 62350080, 62076147, 62061136001, U2341228), Beijing Natural Science Foundation (No. L247011), BNRist (BNR2022RC01006), Tsinghua Institute for Guo Qiang, and the High Performance Computing Center, Tsinghua University. Y. Zhang was supported by the CIE-Tencent PhD Research Incentive Program, and J. Zhu was supported by the XPlorer Prize. This work was done during Yichi Zhang's visit to the van de Schaar lab at the University of Cambridge, which was also supported by the Tsinghua Scholarship for Overseas Graduate Studies.

## Impact Statement

This work tackles reliable verification of AI agents in high-stakes applications such as clinical diagnosis by grounding verification in professional guidelines. Our framework could enhance patient safety by flagging guideline-inconsistent reasoning, provide interpretable signals for expert check, and is potential to generalize to other domains with explicit standards. An important emphasis is that verification should complement rather than replace qualified human judgment, as the signals of correctness are probabilistic and the guidelines themselves may contain flaws or be outdated. Broader validation across diverse healthcare settings might be needed to deploy the method in practice. In summary, this work underscores the value of domain-grounded verification with explicit uncertainty quantification, as validated by expert oversight, for responsible high-stakes AI deployment.

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

# A. Properties of Guideline-Grounded Signals

In this section, we present a more detailed discussion on the desired properties of surrogate evidence in Section 3.2.

## A.1. Error Bound Analysis

We hereby provide a theoretical justification for using simple linear calibration by presenting an error bound for our verification method. Our analysis shows that when guideline-grounded signals satisfy two key properties introduced in Section 3.2, low-capacity linear calibration achieves good generalization even with limited supervision.

We first provide a formal description of the remark. Let $p^* := P(Z = 1|\tau_{1:t})$ denote the true probability that a trajectory leading to answer $y$ is correct, and let $\hat{p}_t$ denote our calibrated estimate for a prefix from GLEAN. We analyze the expected verification error $\mathbb{E}[(\hat{p}_t - p^*)^2]$ under two assumptions about the guideline-grounded surrogate evidence $S_t$ at step $t$. Below, we neglect $t$ for $S_t$ and provide an analysis over arbitrary prefix length.

Let $(\tau, S, Z) \in \mathcal{C}$ be the calibration distribution, where $S \in \mathbb{R}$ is a scalar aggregated signal for trajectory $\tau$, and $Z \in \{0, 1\}$ indicates correctness.

**Assumption A.1** (Approximate Sufficiency). The accumulated surrogate evidence $S$ captures most information about trajectory correctness:

$$\mathbb{E}_{\tau, S \sim \mathcal{C}}[(p_S(S) - p^*(\tau))^2] \leq \epsilon_{\text{suff}} \tag{8}$$

where $p_S(S) = P(Z = 1|S)$ and $\epsilon_{\text{suff}}$ represents the information loss from reducing the trajectory to the accumulated signal.

**Assumption A.2** (Monotone Near-Linearity). Given the accumulated surrogate evidence $S$, there exist $a > 0, c \in \mathbb{R}$ that allow $\forall S$

$$\text{logit}(p_S(S)) = aS + c + \delta(S) \tag{9}$$

where $a > 0$ ensures monotonicity and $\sup_S |\delta(S)| \leq \epsilon_{\text{lin}}$ bounds the deviation from perfect linearity.

Under Assumptions A.1 and A.2, we present a proposition as follows.

**Proposition A.3.** *Let $\mathcal{F} \subseteq \{f : \mathbb{R} \to [0, 1]\}$ be a class of probabilistic calibrators, which contains the linear-logit model $g^*(S) = \sigma(aS + c)$ as introduced in Theorem A.2. Given $M$ calibration samples $\{(S^m, Z^m)\}_{m=1}^M$, define the Brier squared loss risk and its empirical counterpart as:*

$$\mathcal{L}(f) \triangleq \mathbb{E}_{S, Z \sim \mathcal{C}}[(f(S) - Z)^2],$$

$$\hat{\mathcal{L}}_M(f) \triangleq \frac{1}{M} \sum_{m=1}^M (f(S^m) - Z^m)^2,$$

*and let $\hat{f}$ be the empirical risk minimizer (ERM) such that $\hat{f} \triangleq \arg\min_{f \in \mathcal{F}} \hat{\mathcal{L}}_M(f)$.*

*Then, for any $\delta \in (0, 1)$, with a probability at least $1 - \delta$ over the drawn calibration samples,*

$$\mathbb{E}_{\tau, S \sim \mathcal{C}}[(\hat{f}(S) - p^*(\tau))^2] \leq \quad 2\epsilon_{\text{suff}}^2 + \frac{\epsilon_{\text{lin}^2}}{8} + O\left(\sqrt{\frac{Pdim(\mathcal{F}) \log M + \log(1/\delta)}{M}}\right), \tag{10}$$

*where $Pdim(\mathcal{F})$ denotes the pseudo-dimension (Bartlett et al., 2019; Khavari & Rabusseau, 2021), the real-valued capacity measure generalized from VC dimension (Mohri et al., 2018), that reflects the complexity of models.*

*Proof.* For any fixed S and any function $f$, we have

$$\mathbb{E}[(\hat{f}(S) - Z)^2|S] = (\hat{f}(S) - \mathbb{E}[Z|S])^2 + \text{Var}[Z|S] = (\hat{f}(S) - p_S(S))^2 + \text{Var}[Z|S].$$

Taking the expectation over $S$ with the variance being independent of $f$ gives the identity,

$$\mathcal{L}(f) - \mathcal{L}(p_S) = \mathbb{E}[(f(S) - p_S(S))^2]. \tag{11}$$

We define $\Delta_M = \sup_{f \in \mathcal{F}} |\mathcal{L}(f) - \hat{L}_M(f)|$. Then for the ERM $\hat{f}$ learned on the calibration samples, we have

$$\mathcal{L}(f) \leq \hat{\mathcal{L}}_M(f) + \Delta_M, \; \hat{\mathcal{L}}_M(f) \leq \mathcal{L}(f) + \Delta_M, \; \forall f \in \mathcal{F},$$
$$\mathcal{L}(\hat{f}) \leq \inf_{f \in \mathcal{F}} \mathcal{L}(f) + 2\Delta_M.$$

Subtracting $\mathcal{L}(p_S)$ and applying Equation (11) leads to

$$\mathbb{E}[(\hat{f}(S) - p_S(S))^2] \leq \inf_{f \in \mathcal{F}} \mathbb{E}[(f(S) - p_S(S))^2] + 2\Delta_M. \tag{12}$$

Notice that $g^* \in \mathcal{F}$ and the sigmoid function is $\frac{1}{4}$-Lipschitz. This gives, for any $S$,

$$|g^*(S) - p_S(S)| \leq \frac{1}{4}|\delta(S)| \leq \frac{1}{4}\epsilon_{\text{lin}},$$

and further yields

$$\inf_{f \in \mathcal{F}} \mathbb{E}[(f(S) - p_S(S))^2] \leq \mathbb{E}[(g^*(S) - p_S(S))^2] \leq \left(\frac{\epsilon_{\text{lin}}}{4}\right)^2. \tag{13}$$

By the definition of $\Delta_M$ and the generalization bound for bounded, real-valued function (Khavari & Rabusseau, 2021), for any $\delta \in (0, 1)$, with a probability at least $1 - \delta$,

$$\Delta_M \leq O\left(\sqrt{\frac{\text{Pdim}(\mathcal{F}) \log M + \log(1/\delta)}{M}}\right). \tag{14}$$

Therefore, combining Equations (13) and (14) along with the sufficiency assumption, we eventually have

$$\begin{aligned}
&\mathbb{E}[(\hat{f}(S) - p^*(\tau))^2] \\
&\leq 2\mathbb{E}[(\hat{f}(S) - p_S(S))^2] + 2\mathbb{E}[(p_S(S) - p^*(\tau))^2] \\
&\leq 2\epsilon_{\text{suff}}^2 + \frac{\epsilon_{\text{lin}^2}}{8} + O\left(\sqrt{\frac{\text{Pdim}(\mathcal{F}) \log M + \log(1/\delta)}{M}}\right).
\end{aligned}$$

$\square$

**Discussion.** Intuitively, Theorem A.1 states that guideline alignment scores, when accumulated along the trajectory, preserve most information relevant to determining correctness. While individual steps may contain additional details, the accumulated evidence $S_t$ captures the essential information. Empirically, we validate the informativeness of guideline-grounded surrogate evidence by demonstrating their significant discrimination between correct and incorrect trajectories, as shown in Section 3.2 as well as Figure 6. We note that strong discrimination is a necessary but not a sufficient condition for statistical sufficiency. Motivated by the fact that this sufficiency may not hold perfectly, we further introduce active verification strategies to reduce the residual uncertainty not captured by $S_t$.

Meanwhile, Theorem A.2 states that the relationship between accumulated evidence and correctness probability is approximately linear in logit space, which motivates the use of logistic regression for calibration. The monotonicity straightforwardly indicates that better guideline alignment leads to a higher correctness probability. In Section 3.2, we fit a linear model to the logit of probabilities to confirm this property for guideline-grounded evidence. The $R^2$ is 0.943 as reported in Figure 2, which serves as a practical indicator for $\epsilon_{\text{lin}}$. Qualitatively, we also visualize the logit of $P(Z = 1|S_t)$ against $S_t$, where near-linearity manifests as an approximately straight line.

When these two properties approximately hold, indicating that the two approximation terms tend to be small, the remaining error is primarily governed by the capacity-controlled generalization error in terms of the model family. Since a one-dimensional or low-dimensional linear calibrator has a substantially smaller capacity than more expressive models (Mohri et al., 2018), it would achieve the same calibration error with fewer calibration samples. This supports our choice of a linear calibrator under the scarce-supervision regime which is typical in high-stakes settings.

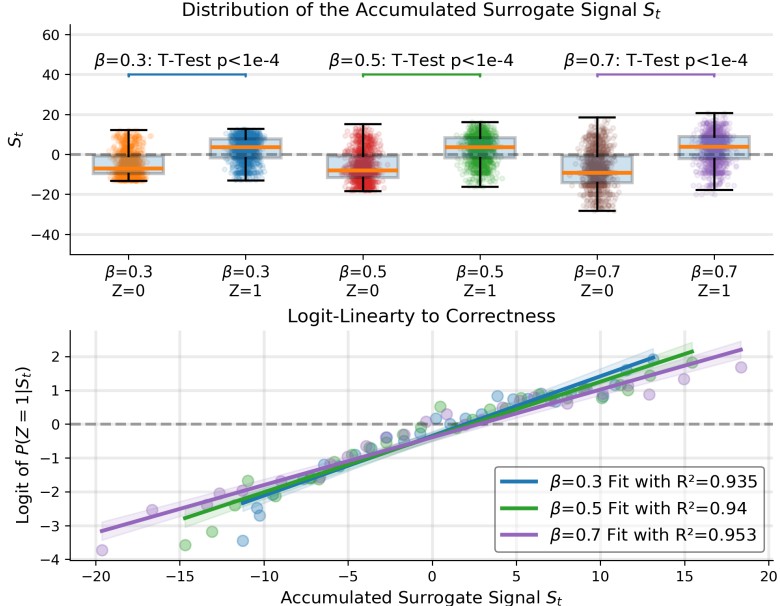

*Figure 6.* Properties of guideline-grounded signals accumulated with $\beta$-discounting.

## A.2. Properties Validation

As our accumulated surrogate evidence is practically implemented as a $\beta$-discounted sum over step-level alignment signals, we additionally verify that the resulting $S_t^{(\beta)} = \sum_{i=1}^{t} \beta^{t-i} \log \frac{s_i}{1-s_i}$ preserves the desired properties across a range of discount factors. Specifically, we construct prefix-level surrogate evidence with $\beta \in \{0.3, 0.5, 0.7\}$, and examine whether it remains informative for correctness. As shown in the top panel of Figure 6, the distributions of $S_t^{(\beta)}$ for correct and incorrect trajectories are consistently well separated for all $\beta$, with statistically significant mean differences (t-tests, $p < 1e - 4$). This indicates that discounting does not destroy the discriminative signal of the accumulated evidence, and that $S_t^{(\beta)}$ remains an informative, compact summary of guideline-grounded trajectory quality.

We further assess the monotone near-linearity assumption in logit space for discounted accumulation. For each $\beta$, we estimate $P(Z = 1 | S_t^{(\beta)})$ via binning and compute its logit, then fit a one-dimensional linear model. The bottom panel of Figure 6 shows that the logit-transformed correctness probability aligns closely with a linear trend over $S_t^{(\beta)}$, resulting in high $R^2 \approx 0.94 - 0.95$ across different discount factors. Notably, the fitted slopes are similar across $\beta$, suggesting that changes in the discount factor do not impact the fundamental relationship to correctness, but only adjust the precision with a few different information. This robustness implies that the proposed properties being approximately satisfied mainly arise from guideline grounding via accumulation rather than the hyperparameter selection, and supports our choice to use a single linear calibrator in practice.

## B. Details of GLEAN

We first provide the algorithm of GLEAN in Algorithm 1 and then respectively introduce more implementation details and additional experimental results.

### B.1. Implementation Details

**Baselines.** For model-based ratings, we provide the model with the input and output of the diagnosis from the agent and ask it to decide whether the diagnosis is correct or to provide its confidence in the correctness. Specifically, we ask the model to return YES/NO for $P(\texttt{True})$ and a scalar score from 0 to 1 for LLM-as-a-Judge. For sampling-based models, it is difficult to group the answers directly due to the open-ended nature of clinical diagnosis. We therefore embed the candidate answers with an embedding model and calculate their embedding similarities with the answer to be verified. We consider them to be the same if the similarity is higher than 0.6 and calculate the ratio of supporting candidates for self-consistency. We cluster the candidates using the same criteria to calculate the semantic entropy. As for self-verification, we ask the model to decide

---

**Algorithm 1** Guideline-Grounded Evidence Accumulation

---

**Input:** trajectory $\tau_{1:T} = \{o_t, a_t\}_{t=1}^T$, answer $y$, judge $\mathcal{J}$, guideline set $\mathcal{G}$, calibration parameters $\mathbf{w}, b \sim p(\cdot|\mathcal{D})$, discount factor $\beta$, threshold $\epsilon_u$, rectification factor $\alpha$

**Output:** calibrated verification signal for answer $\hat{p}_T$

Retrieve relevant guidelines $\hat{\mathcal{G}} \subset \mathcal{G}$ for answer $y$

**for** $t = 1$ to $T$ **do**

  // *Multi-Guideline Aggregation*

  **for** each guideline $g \in \hat{\mathcal{G}}$ **do**

    Query $\mathcal{J}$ whether $(o_t, a_t)$ aligns $g$ to get $s_{t,g} \in [0,1]$

  **end for**

  Aggregate: $\mathbf{s}_t \leftarrow \Phi(\{s_{t,g}\}_{g \in \hat{\mathcal{G}}})$

**end for**

Accumulate: $\mathbf{S}_T \leftarrow \sum_{t=1}^T \beta^{T-t} \log \frac{\mathbf{s}_t}{1-\mathbf{s}_t}$

Calibration: $\hat{p}_T \leftarrow \mathbb{E}_{\mathbf{w},b \sim p(\cdot|\mathcal{D})}[\sigma(\mathbf{w}^\top \mathbf{S}_T + b)]$

**if** $H(\hat{p}_T) > \epsilon_u$ **then**

  // *Guideline Expansion*

  Retrieve more guidelines $\hat{\mathcal{G}}^+ \subset \mathcal{G}$ for answer $y$

  $\hat{\mathcal{G}} \leftarrow \hat{\mathcal{G}} \cup \hat{\mathcal{G}}^+$

  Query $\mathcal{J}$ for each $g \in \hat{\mathcal{G}}^+$ and get $\{s_{t,g}\}_{g \in \hat{\mathcal{G}}}$

  // *Differential Checks*

  Retrieve competitive guidelines $\hat{\mathcal{G}}^- \subset \mathcal{G}$ for $y' \neq y$

  Query $\mathcal{J}$ for each $g \in \hat{\mathcal{G}}^-$ and get $\{s_{t,g}\}_{g \in \hat{\mathcal{G}}^-}$

  Correct $\{s_{t,g}^{rec}\}_{g \in \hat{\mathcal{G}}}$ with Equation (7)

  Recompute $\mathbf{S}_T$ and estimate $\hat{p}_T$

**end if**

---

whether each observation complies with the typical symptoms of the diagnosis and take the average score across each check. We supplement $P(\texttt{True})$ with the diagnosis process and the retrieved guideline to implement RAG-augmented verification. We follow the practice of Med-PRM (Yun et al., 2025) for methods with reward models, which trains the model to output two special tokens and calculate the likelihood of the positive one as the reward.

**Guideline Retrieval.** To retrieve guidelines for different predictions, we first prompt an LLM, i.e., GPT-4o, to extract the primary diagnosis from the raw prediction. We take these extracted diagnosis terms and frequent words for clinical diagnosis (e.g., "management", "diagnosis") as the keywords and run keyword matching against the titles of guidelines to calculate the lexical overlap at the word level. If multiple titles have a same number of keywords matched, then we use an embedding model, `all-mpnet-base-v2`, to rank them. After that, we take top-K guidelines to support the verification.

**Step-wise Rating.** Specifically, we prompt the agent backbone with the history up to the current step and ask it to decide whether the observation and the action align with the provided guideline in the manner of $P(\texttt{True})$ to save output tokens. We use the following prompt for guideline-grounded step-wise rating and minimal changes are made for ablations. We get the top-10 logits from the model for the first token and calculate the token probability following Equation (3).

---

**Prompt for Guideline-grounded Step-wise Rating**

You are a board-certified clinician and a strict guideline compliance evaluator.

You will be given: (1) The diagnosis history so far (prior steps),
(2) The current step, including the new observation(s) and the decision/action taken,
(3) A reference clinical guideline relevant to the proposed diagnosis or management.

Your job is NOT to judge the entire final diagnosis. Your job is to judge whether the CURRENT STEP is clinically appropriate and consistent with the guideline, given the available information up to this step.

---

Be strict and conservative:
- Answer YES only if the current step is clearly supported by the patient information so far AND is consistent with the guideline.
- Answer NO if the step is unsupported, premature, contradicts key facts, violates guideline recommendations, skips required checks, or is not justified as the next best step.

Reply with exactly one token: YES or NO.

Diagnosis so far (prior steps):
{history}

Current step:
Observation(s):
{observation}

Rationale & Action:
{action}

Reference guideline:
{guideline}

Task: Is the CURRENT STEP appropriate and guideline-consistent given the patient information so far?
Reply with YES or NO only.

**Active Verification.** For guideline expansion, we take the highest-ranking guideline that has not been used previously for additional judgment. As for differential checks, we first extract competitive or differential conditions for the predicted diagnosis. Instead of asking an LLM, we seek similar ones in the generated diagnosis for the specific disease and randomly pick two guidelines from them. Considering that the diagnosed conditions are given based on patients with the same disease, the candidates are highly likely to be competing alternatives.

*Table 5.* **Ablations on aggregation strategies.**

| Φ | AUROC | Risk | ECE | Brier |
|---|---|---|---|---|
| Agent Backbone: *Qwen2.5-7B-Instruct* | | | | |
| avg | 0.8870 | 0.1767 | 0.0643 | 0.1383 |
| min | 0.9114 | 0.1656 | 0.0362 | 0.1202 |
| avg, min | 0.9138 | 0.1678 | 0.0469 | 0.1181 |
| avg, min, max, std | 0.9130 | 0.1644 | 0.0427 | 0.1193 |
| Agent Backbone: *Qwen3-30B-A3B-Instruct* | | | | |
| avg | 0.9025 | 0.1767 | 0.1088 | 0.1382 |
| min | 0.9385 | 0.1278 | 0.0595 | 0.0976 |
| avg, min | 0.9389 | 0.1300 | 0.0700 | 0.0947 |
| avg, min, max, std | 0.9387 | 0.1256 | 0.0598 | 0.0983 |

*Table 6.* **Ablations on the selection of $\beta$.**

| $\beta$ | AUROC | Risk | ECE | Brier |
|---|---|---|---|---|
| Agent Backbone: *Qwen2.5-7B-Instruct* | | | | |
| 0.1 | 0.8988 | 0.1844 | 0.0635 | 0.1334 |
| 0.3 | 0.9083 | 0.1767 | 0.0538 | 0.1244 |
| 0.5 | 0.9137 | 0.1678 | 0.0418 | 0.1178 |
| 0.7 | 0.9145 | 0.1678 | 0.0352 | 0.1154 |
| 0.9 | 0.9080 | 0.1678 | 0.0314 | 0.1189 |
| Agent Backbone: *Qwen3-30B-A3B-Instruct* | | | | |
| 0.1 | 0.9357 | 0.1389 | 0.0800 | 0.1076 |
| 0.3 | 0.9391 | 0.1267 | 0.0602 | 0.0988 |
| 0.5 | 0.9389 | 0.1300 | 0.0700 | 0.0947 |
| 0.7 | 0.9328 | 0.1322 | 0.0420 | 0.0984 |
| 0.9 | 0.9245 | 0.1378 | 0.0330 | 0.1060 |

## B.2. Additional Results

In this section, we present additional experimental results to further justify the design of GLEAN.

**Aggregation Strategy.** In the main text, we adopt $\Phi(\{s_{t,g}\}) = [\min, \text{avg}]_{g \in \hat{\mathcal{G}}} s_{t,g}$, resulting in a 2-dimensional feature for each step. Table 5 presents an ablation on different aggregation functions for combining multi-guideline ratings. We compare four strategies, including averaging (avg), taking the minimum (min), combining both (avg, min), and using comprehensive statistics (avg, min, max, std). Across both agent backbones, we observe that conservative aggregation via the minimum rating performs strongly, achieving AUROC of 0.9114 and 0.9385 for the two backbones respectively, with notably low calibration error. This suggests that the most critical guideline, the one with the lowest alignment, effectively captures deviations from acceptable reasoning. Combining average and minimum provides a slight improvement, balancing conservative assessment with overall alignment trends (AUROC 0.9138 and 0.9389). Interestingly, adding more statistics, i.e., maximum and standard deviation, does not consistently improve performance and sometimes increases calibration error, likely due to noises in the low-data regime. Based on these results, we adopt (avg, min) as our default aggregation strategy.

**Hyperparameter Selection.** We ablate the discount factor $\beta$ (Table 6) and the rectification factor $\alpha$ (Table 7). For $\beta$, with $\beta = 0.1$ over-emphasizing recent evidence and $\beta = 0.9$ giving excessive weight to noisy early steps, these extreme values underperform. Moderate values ($\beta \in [0.3, 0.7]$) achieve strong performance. We select $\beta = 0.5$, which balances down-weighting noisy early signals while incorporating trajectory history, achieving AUROC of 0.9137 and 0.9389 with good calibration (ECE 0.0418 and 0.0700). For $\alpha$ in differential checks, we observe a discrimination-calibration trade-off. Although $\alpha = 0.5$ achieves the highest AUROC, it significantly degrades calibration, especially for Qwen3-30B. Since well-calibrated confidence is critical for trustworthy high-stakes verification, we select $\alpha = 0.2$, which maintains strong discrimination with reliable probability estimates. This corresponds to a conservative rectification that incorporates counter-evidence without over-penalization.

*Table 7.* **Ablations on the selection of $\alpha$.**

| $\alpha$ | AUROC | Risk | ECE | Brier |
|---|---|---|---|---|
| Agent Backbone: *Qwen2.5-7B-Instruct* | | | | |
| 0.1 | 0.9411 | 0.1344 | 0.0639 | 0.0989 |
| 0.2 | 0.9470 | 0.1300 | 0.0700 | 0.0947 |
| 0.5 | 0.9480 | 0.1256 | 0.0777 | 0.1009 |
| Agent Backbone: *Qwen3-30B-A3B-Instruct* | | | | |
| 0.1 | 0.9518 | 0.1133 | 0.0607 | 0.0864 |
| 0.2 | 0.9557 | 0.1078 | 0.0653 | 0.0849 |
| 0.5 | 0.9601 | 0.1033 | 0.1113 | 0.0984 |

*Table 8.* **Analysis of computational costs** with $N = 16$ for generation budget, $T$ for execution steps, $K$ for guidelines to use.

| Method | Complex. | #LLM Calls | #Tokens(k) | AUROC | Brier |
|---|---|---|---|---|---|
| Agent Backbone: *Qwen2.5-7B-Instruct* | | | | | |
| Self-Consistency | $O(T \cdot N)$ | 78.3 | 135.2 | 0.8368 | 0.1886 |
| GLEAN ($K = 1$) | $O(T)$ | 4.7 | 22.1 | 0.8574 | 0.1554 |
| GLEAN ($K = 3$) | $O(T \cdot K)$ | 14.0 | 78.0 | 0.9137 | 0.1178 |
| +Active ($\epsilon_u = 0.5$) | $O(T \cdot K)$ | 20.3 | 132.7 | 0.9345 | 0.0992 |

**Computational Costs.** We analyze the computational costs of GLEAN and compare it with the best-performing baseline, Self-Consistency, in Table 8. Let $T$ denote trajectory length, $K$ the number of guidelines, and $N$ candidate samples. Self-consistency has a complexity of $O(N \cdot T)$, requiring $N$ full-trajectory generations, while GLEAN has a complexity of $O(T \cdot K)$, verifying a single trajectory with $K$ guideline-grounded ratings per step, where the rating extraction via token probabilities is much cheaper than autoregressive generation. Empirically, self-consistency uses 78.3 LLM calls and 135.2k tokens to achieve AUROC of 0.8368 with 16 generations for each case, while GLEAN ($K = 1$) delivers better performance with only 4.7 calls and 22.1k tokens. At $K = 3$, GLEAN achieves AUROC of 0.9137 with 78.0k tokens, substantially outperforming self-consistency with a lower cost, and active verification adds moderate overhead but further improves discrimination and calibration. These results demonstrate that guideline grounding provides reasonable performance-cost trade-offs, especially compared to test-time scaling methods in generation.

**Cross-Agent Verification.** Table 9 examines cross-backbone verification. We take the verifiers individually trained with different agent backbones and evaluate their verification performance on trajectories by the other agent. From the results, we can see that cross-agent verification remains consistently competitive with reasonable discrimination. Qwen3-30B, as a stronger model, yields better verification signal than Qwen2.5-7B on both agents. Yet, the calibration for Qwen3-30B when verifying the other agent degrades. This indicates that while guideline-grounded signals generalize across agents, the

calibration for evidence can be partially agent-specific, since the signals may contain specific patterns for the agent. Table 10 further confirms the stability across agent backbones. We find that active verification remains beneficial overall, with most metrics improved across verifier backbones. Even with Qwen2.5-7B as a weaker model, we can see improvements with it as the verifier instead of degradation.

*Table 9.* **Cross verification across two agent backbones.**

| Verifier | Agent | | | | | | | |
|---|---|---|---|---|---|---|---|---|
| | Qwen2.5-7B | | | | Qwen3-30B | | | |
| | AUROC | Risk@0.5 | ECE | Brier | AUROC | Risk@0.5 | ECE | Brier |
| **Qwen2.5-7B** | 0.9137 | 0.1678 | 0.0418 | 0.1178 | 0.9056 | 0.1667 | 0.0310 | 0.1209 |
| **Qwen3-30B** | 0.9200 | 0.1633 | 0.1165 | 0.1368 | 0.9389 | 0.1267 | 0.0550 | 0.0975 |

*Table 10.* Active verification with judges based on different LLMs by extending the results in Table 9 to active one.

| Verifier | Agent | AUROC | Risk0.5 | ECE | Brier |
|---|---|---|---|---|---|
| Qwen3 | Qwen2.5 | $0.9200 \rightarrow 0.9475$ | $0.1633 \rightarrow 0.1278$ | $0.1352 \rightarrow 0.1276$ | $0.1368 \rightarrow 0.1256$ |
| Qwen2.5 | Qwen3 | $0.9056 \rightarrow 0.9163$ | $0.1667 \rightarrow 0.1622$ | $0.0310 \rightarrow 0.0356$ | $0.1209 \rightarrow 0.1151$ |

**Detailed results for Best-of-N.** We also include the detailed results for performance with Best-of-N in Table 11, which correspond to Figure 4.

*Table 11.* **Accuracy with Best-of-N selection.**

| Method | Qwen2.5-7B (Pass@1: 52.16%) | | | Qwen3-30B (Pass@1: 58.94%) | | |
|---|---|---|---|---|---|---|
| | BoN-4 | BoN-8 | BoN-16 | BoN-4 | BoN-8 | BoN-16 |
| P(True) | 0.5386 | 0.5402 | 0.5380 | 0.6322 | 0.6245 | 0.6133 |
| LLM-as-a-Judge | 0.5818 | 0.6062 | 0.6200 | 0.6424 | 0.6396 | 0.6467 |
| Self-Consistency | 0.6209 | 0.6617 | 0.6940 | 0.6525 | 0.6865 | 0.7000 |
| RAG | 0.6891 | 0.7140 | 0.7240 | 0.6901 | 0.6933 | 0.6900 |
| Self-Verification | 0.6366 | 0.6474 | 0.6440 | 0.6743 | 0.6617 | 0.6200 |
| MedPRM | 0.5723 | 0.5777 | 0.5840 | 0.6468 | 0.6582 | 0.6600 |
| ORM | 0.5965 | 0.6159 | 0.6440 | 0.6801 | 0.6999 | 0.6966 |
| GLEAN (N=1) | 0.6996 | 0.7301 | 0.7360 | 0.7198 | 0.7237 | 0.7167 |
| GLEAN (N=3) | 0.7052 | 0.7429 | 0.7660 | 0.7383 | 0.7671 | 0.7833 |

## B.3. Expert Study Design

To validate the clinical utility and interpretability of the GLEAN verification framework, we developed a custom evaluation interface and conducted an expert study with three certified clinicians. The three clinicians were selected from three countries across two continents, providing geographic diversity in the evaluation. They have over five years of clinical experience on average. Two are primary care physicians, and one is an MD-PhD with experience spanning clinical medicine and academia. All three regularly encounter the studied diseases in diagnosis and are also familiar with AI. Let us now provide further details on the study design.

### B.3.1. TRAJECTORY SELECTION

We selected **50 diagnostic trajectories** from our evaluation set, ensuring diversity across:

- **Diseases**: Pancreatitis, cholecystitis, and diverticulitis

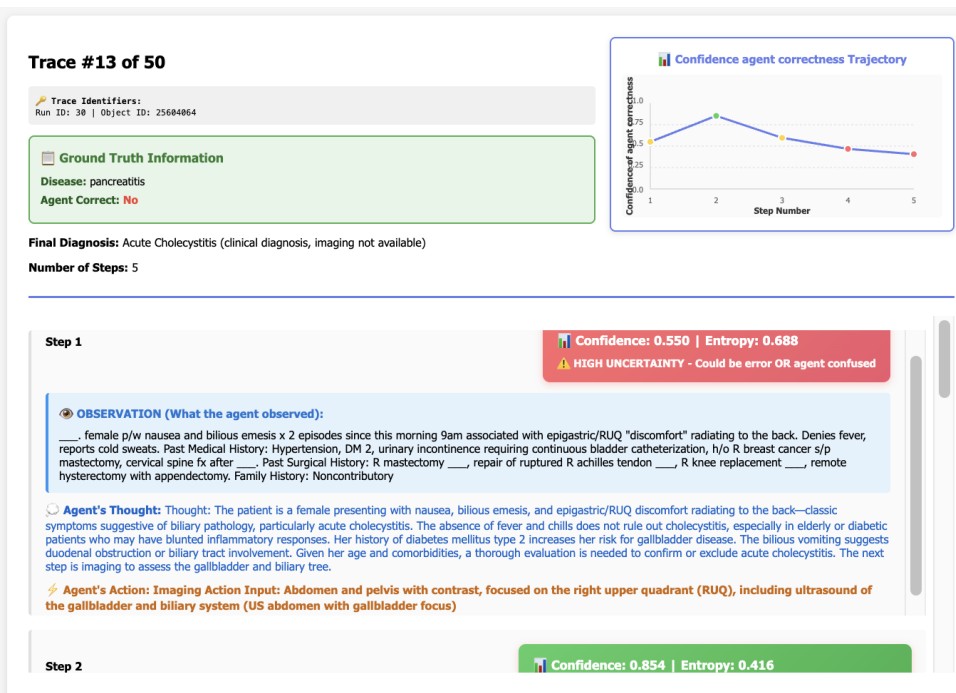

*Figure 7.* **Annotation interface shown to clinicians showing the step by step agent trajectory (scroll bar) and GLEAN metrics.**

- **Agent correctness**: Balanced mix of correct and incorrect final diagnoses

- **Trajectory length**: Varying numbers of reasoning steps (3–8 steps)

- **GLEAN confidence patterns**: Mix of high-confidence, low-confidence, and trajectories with large confidence shifts

This stratified sampling ensured coverage of different verification scenarios that GLEAN might encounter in practice.

### B.3.2. DISPLAYED INFORMATION PER TRACE.

Before beginning evaluations, clinicians received written instructions: (1) The purpose of the study, (2) How GLEAN's verification metrics are computed (at a high level), (3) Interpretation of confidence scores and uncertainty, (4) How to use the evaluation interface.

Then, during evaluation for each trace, we have a lightweight HTML web interface that shows the full step-by-step agent trajectory (see Figure 7). The interface displayed the trajectory sequentially. For each step, the reviewer viewed the Agent's Observation (clinical data), Thought (reasoning trace), and Action (tool use), alongside the calculated GLEAN metrics per step, namely Confidence of correctness and Uncertainty (entropy).

### B.3.3. QUESTIONNAIRE AND SCORING RUBRIC

For each of the 50 traces, clinicians answered a structured questionnaire. The specific questions and exact score ranges used in the study were as follows:

**Localization**

1. **First Error Step:** *At which step did the agent first make a diagnostic error?* (Input: Step Number or "None")

2. **Breakthrough Step:** *At which step did the agent make the KEY correct decision that led to the right diagnosis?* (Input: Step Number or "N/A")

**Qualitative Assessment (Likert Scales)** Participants rated the method using the following exact 5-point scales:

Extremely easy

▶ *Interpretability*: *How easy was it to use the metrics to understand the agent's reliability?* **1:** Very confusing; **2:** Somewhat confusing; **3:** Moderately clear; **4:** Very clear; **5:** Extremely clear

▶ *Clinical Utility*: *How useful was this verification method for reviewing this trace?* **1:** Not useful; **2:** Slightly useful; **3:** Moderately useful; **4:** Very useful; **5:** Extremely useful

### B.3.4. METRICS

These results across traces are then collated across clinicians, and the metrics are computed as per the main paper.

