# OpenReview forum: "GLEAN: Guideline-Grounded Evidence Accumulation for High-Stakes Agent Verification"
_ICML.cc/2026/Conference — ICML 2026 regular_

### Official Review · Reviewer_FtdR · 2026-02-22

**Soundness:** 2
**Presentation:** 2
**Significance:** 2
**Originality:** 1
**Overall Recommendation:** 2
**Confidence:** 3

**Summary:**

This paper studies verification for LLM-based agents in high-stakes decision-making, with clinical diagnosis as the main testbed. The authors argue that existing verifiers often fail due to insufficient domain grounding and poor calibration, and propose GLEAN as a verification framework that turns expert-curated guidelines into trajectory-aware and calibrated correctness signals.

**Compliance With Llm Reviewing Policy:**

Affirmed.

**Final Justification:**

After considering the quality and novelty of the paper, as well as the other reviewers’ comments and the authors’ rebuttal, I maintain my initial assessment. In particular, the novelty of this paper does not meet my threshold for ICML. As I noted earlier, Med-PRM appears to share a similar motivation with GLEAN, whereas the rebuttal characterizes it as only superficially related. However, the authors also include Med-PRM as a baseline, which further suggests meaningful overlap rather than mere superficial similarity. More broadly, related design elements can also be found in other prior works. Therefore, at this stage, I keep my current rating.

**Key Questions For Authors:**

1. Can you quantify performance vs. retrieval quality? If performance collapses under mild noise, the method may be retrieval-bottlenecked, while if it degrades gracefully, that strengthens the practical claim.
2. It would strengthen the paper if the authors justified the validity of using an LLM’s YES/NO token probabilities as a step-level rating signal.

**Limitations:**

yes

**Strengths And Weaknesses:**

**Strengths**
1.  The derivation from sequential log-odds accumulation to a surrogate evidence signal is coherent and gives a principled backbone for trajectory-level verification rather than single-shot judging.
2. Using Bayesian logistic regression and posterior sampling is a concrete step beyond many confidence heuristics.

**Weaknesses**
1. The method’s raw step rating relies on a specific prompting. These can shift substantially across models and templates.
2. Since supervision is at the trajectory level, step-level alignment could be punished even when it is clinically reasonable but the final label is wrong (or vice versa).
3. Symbols like $o_t$ and $a_t$ are introduced midway through Section 3.1 and can be easy to miss. Although their meaning is largely inferable from context, I recommend that the authors define them explicitly.
4. Individual components are not individually new.

---

> ### Author Rebuttal · Authors · 2026-03-31
>
> We thank the reviewer for the feedback. To respond clearly, we address your concerns from four points.
>
> Our new results are in https://i.ibb.co/x4hBGyJ/R4.png
>
> **1. Novelty of the overall framework**
>
> We respectfully disagree with the assessment that the paper lacks originality simply because each component is not individually new. Notably, the other reviewers consistently viewed the paper as original, especially for the **_sequential formulation, active verification, and differential check_**. Our novelty lies in introducing a new, principled verification framework for high-stakes agents with guideline-grounded sequential evidence accumulation, calibrated into correctness probabilities and extended with uncertainty-triggered active verification.
>
> More concretely, our novelty is threefold.
> * First, conceptually, we reformulate **agent verification as sequential evidence accumulation grounded in domain knowledge** via professional guidelines. To our knowledge, this framework is not only new in agent verification, but also of great significance for trustworthy real-world deployment, with clinical diagnosis as a representative.
> * Second, we show that **guideline-grounded step ratings exhibit approximate sufficiency and monotonic near-linearity with correctness**, whereas unguided ratings do not (**Figure 2**). This directly justifies our surrogate-evidence design and the calibrated evidence-accumulation pipeline. This does not follow trivially from the individual components alone.
> * Third, we introduce **uncertainty-triggered active verification**, linking verification to test-time scaling in a domain-grounded way by selectively expanding guideline coverage and incorporating counter-evidence from competitive alternatives.
>
> We politely ask the reviewer to reassess the originality of the paper, especially at the level of the overall framework, regarding its practical, conceptual, and technical significance..
>
> **2. Performance v.s. Retrieval Quality**
>
> Thanks for pointing this out. We recognize imperfect retrieval as a realistic case, and active verification is explicitly designed to address it by expanding guideline coverage and introducing counter-evidence when the initial evidence is uncertain. To study this, we conduct a controlled retrieval-corruption experiment, where a fraction of originally retrieved high-quality guidelines is replaced with less relevant ones (**Table 1 in the link**). We find that GLEAN degrades gracefully rather than collapsing abruptly as retrieval quality worsens, and that active verification recovers a substantial portion of the loss. Meanwhile, calibration remains broadly stable, with only small changes in ECE and Brier. Overall, these results support active verification as a practical mechanism for mitigating imperfect retrieval while maintaining a robust calibrator.
>
> **3. Validity of step rating**
>
> We address the concerns regarding step-level rating with three points below.
> * _YES/NO token probabilities as signal:_ Using token probabilities over a discrete set is already a common practice in model-based verification, validated by Anthropic[1] and followed by later works[2]. We use the normalized likelihood over {YES,NO}, rather than free-form judgments, so the signal reflects relative support between two explicit answers and is less affected by format-following. Moreover, Section 3.2 directly justifies the informativeness of this signal, where guideline-grounded step ratings exhibit approximate sufficiency and strong monotonic near-linearity with correctness. The main experiments also confirm its effectiveness.
> * _Shift in models and templates:_ We empirically show that the overall performance of GLEAN is stable across different models and prompts (**Table 2,3 in the link**). First, the same prompting scheme works effectively across multiple backbones in our experiments, and our cross-agent results further show that the framework remains beneficial under judge-model shifts. Second, we also tested several prompt variants that modify the instruction and guideline presentation. While absolute metrics vary slightly, the behavior is qualitatively unchanged and remains much better than baselines.
> * _Reasonable steps with wrong prediction:_ We would like to clarify that this is precisely the target we want to identify with our framework. GLEAN does not assume that every step in an incorrect trajectory must be wrong. Instead, it asks whether the observed trajectory provides sufficient support for the final answer under guidelines. When the diagnosis is wrong, the retrieved guidelines are more likely to conflict with internal steps, reducing the accumulated support and thus the final calibrated confidence (also validated in R.qmRm-Table 2). We will clarify this in the revision.
>
> [1] Kadavath et al., Language Models (Mostly) Know What They Know
>
> [2] Kapoor et al., Large Language Models Must Be Taught to Know What They Don't Know
>
> **4. Notation**
>
> We will refine the exposition in revision.

---

> > ### Author Rebuttal · Reviewer_FtdR · 2026-04-03
> >
> > The authors’ rebuttal and additional experiments are appreciated, and I have carefully read the comments from the other reviewers. I agree that the paper presents a coherent framework, but I still find the conceptual novelty limited. In my view, the main contribution lies in the combination of guideline-grounded stepwise scoring, evidence accumulation, calibration, and active verification. In particular, I am not convinced by the claim that the framework is ''new in agent verification." Prior work such as ''Med-PRM: Medical Reasoning Models with Stepwise, Guideline-verified Process Rewards" has already explored guideline-grounded stepwise verification in clinical diagnosis. Therefore, while your formulation may be more explicit in casting the problem as sequential evidence accumulation, the underlying conceptual ingredients appear substantially related to prior work, so this statement seems overstated. In addition, I do not think the rebuttal fully addresses the supervision mismatch issue. My concern is not merely that incorrect trajectories may contain reasonable steps, but that the calibration target is defined only at the trajectory level. As a result, it remains unclear whether the learned signal faithfully captures step-level support, rather than simply correlating with final correctness. Therefore, I maintain my current rating.

---

> > > ### Author Response · Authors · 2026-04-03
> > >
> > > We thank the reviewer for the follow-up comments, and are glad to see our rebuttal has addressed some of your concerns. Below are our responses to the follow-up questions.
> > >
> > > ## Regarding Novelty
> > >
> > > We respectfully disagree with the implication that the conceptual contribution of GLEAN is limited and prior work such as Med-PRM overlaps with our contribution. While Med-PRM is related at a high level in using guideline-grounded stepwise signals, the problem setting and methodology contribution are fundamentally different. Med-PRM is a trained process-reward model for medical reasoning, relying on substantial human annotation, whereas GLEAN is a training-free verification framework for LLM agents that operates at the trajectory level, produces calibrated correctness probabilities, and further supports uncertainty-triggered active verification at test time. These are not minor implementation differences, but differences in both the target problem and the core methodology.
> > >
> > > Moreover, this distinction is also reflected empirically. We have already included Med-PRM as a baseline in the paper, and GLEAN substantially outperforms it. This is important evidence that our contribution is not reducible to prior guideline-grounded stepwise scoring alone. Rather, the trained PRM with massive labeling efforts falls short in generalization, and the sequential accumulation, calibration, and active verification components essentially improve the effectiveness of the system. In particular, Med-PRM does not provide the trajectory-level calibrated verification signal or the active evidence-collection mechanism that are central to GLEAN.
> > >
> > > Therefore, our claim is not that no prior work has ever used guideline-grounded stepwise signals, but that GLEAN introduces a new verification framework for high-stakes LLM agents, which involves guideline-grounded sequential evidence accumulation, calibrated into correctness probabilities and extended with uncertainty-triggered active verification. These technical components are exactly the same as in the reviewer’s conclusion. Thus, we can reach a consensus on the contributions.
> > >
> > > Our contributions, especially on the work position and the technical points, are in fact highly recognized by other reviewers. Reviewer Z7k3 wrote that “_The stepwise scoring, trajectory-level evidence accumulation, and final calibration fit together well, and the active-verification component is a sensible extension._” Reviewer qmRm specifically highlighted “_reframing agent verification as sequential, discounted evidence accumulation_” and “_triggering test-time scaling based on calibrated uncertainty_”, and concluded his assessment on GLEAN as “_a novel and creative synthesis of existing techniques_” and “_highly practical for high-stakes domains_”. Reviewer 8jUp likewise stated that “_GLEAN is generally novel_”, especially for its “_uncertainty triggered verification mechanisms_”.
> > >
> > > For these reasons, we respectfully maintain that the paper’s originality should be assessed at the level of the overall framework and problem setting, rather than by isolating one superficially related ingredient.
> > >
> > > ## Regarding Supervision
> > >
> > > We would like to first clarify that GLEAN does not claim to recover ground-truth step correctness. Instead, the step-wise score is designed as a surrogate evidence signal, intended to reflect how much each step supports or weakens the correctness of the claimed final answer. In this sense, GLEAN should be understood as a form of hypothesis-conditioned verification – given a proposed final answer, each step is evaluated by asking whether it provides support for that hypothesis under the retrieved guidelines. This also indicates that the calibration target in GLEAN is not only defined at the trajectory level, but closely related to the evidence provided to support the final correctness, which is exactly our verification goal.
> > >
> > > The relevant question, therefore, is not whether the score is a perfect step-level label, but whether it is a useful and faithful surrogate of step-level support. We provide two pieces of evidence for this. First, **Figure 2** shows that, once grounded in guidelines, the accumulated step ratings exhibit approximate sufficiency and strong monotonic near-linearity with correctness, whereas unguided ratings do not. Second, our expert study in **Section 4.3 (line 414)** provides complementary human validation that clinicians used GLEAN’s confidence signals to identify the first erroneous step and showed substantial agreement ($\kappa=0.78$) on error identification, suggesting that the learned signal is not merely an arbitrary correlate of final correctness, but indeed helps track where support for the final answer begins to break down along the trajectory.
> > >
> > > We will clarify this point in the revision and make explicit that our claim is about surrogate evidence for trajectory-level verification, rather than recovery of ground-truth step-level correctness.

---

### Official Review · Reviewer_8jUp · 2026-03-10

**Soundness:** 2
**Presentation:** 1
**Significance:** 3
**Originality:** 3
**Overall Recommendation:** 4
**Confidence:** 2

**Summary:**

This work introduced GLEAN, a LLM verification framework for the deployment of agents in high stakes domains. Given the prediction from an agent, GLEAN produces a probability that the given answer was correct. This probability is produced by accumulating evidence iteratively: at each step, a new guideline document is retrieved and a judge agent is used to estimate the marginal change in likelihood that the answer is correct given this new guideline. Through experiments with 3 different prediction tasks from the MIMIC-IV dataset, GLEAN is shown to produce accurate and well calibrated judgements on the correctness of answers.

**Compliance With Llm Reviewing Policy:**

Affirmed.

**Final Justification:**

In my initial review, I was concerned about the limited scope of evaluation, and the fact that several details seemed to be missing from the paper. The authors adequately addressed both of these concerns during the rebuttal by 1) adding many missing details and 2) adding a substantial amount of additional experimental results (new dataset, additional LLM). With these changes, I believe this work is ready for acceptance.

**Key Questions For Authors:**

Please see the questions listed under "weaknesses -- presentation". Clarifying each of these points in the rebuttal would be helpful, but of course clarifying them in the paper is the primary goal.

**Limitations:**

Yes

**Strengths And Weaknesses:**

Strengths:
- Significance
  - The problem studied in this work -- judging whether LLM predictions are correct using external evidence -- is interesting and well motivated. While the work only studies one dataset, the framework should be broadly applicable to many medical applications of agents. The experimental results in this work show a strong improvement in performance over prior work.
- Originality
  - To my knowledge, GLEAN is generally novel. While using incremental evidence from external documents is not particularly new, the uncertainty triggered verification mechanisms are interesting and novel. In particular, gathering contradictory documents in cases with insufficient certainty is an interesting idea.

Weaknesses
  - Presentation
    - Several important details are missing or unclear from this work. I enumerate some specifics around this point below.
      1) It is unclear how the new evidence/action at timestep t are actually selected. Are these informed by previous time steps in any way?
      2) What does it mean to “retrieve competitive outcomes”? Are these different trajectories (and their respective guidelines) that yielded the opposite answer?
      3) What does it mean to “retrieve competitive outcomes”? Are these different trajectories (and their respective guidelines) that yielded the opposite answer?
      4) Are the trajectories used to train the calibration model separate from those used for evaluation? If so, please specify this.
      5) How are the keywords used for guideline extraction determined? It is stated that retrieval is based on keyword matching against document titles, but the source of the keywords is never described.
      6) What was the background of the three expert clinicians? What subspecialty are they, how many years of experience do they have, and are they still practicing?
    - Some of the exposition is fairly confusing, and could be improved. In particular, the description of Bayesian logistic regression in this work is somewhat odd. As written, it is not clear that N(0, lambda^-1) is a prior. Rather, it reads as if these are the posterior distributions over which the expected parameter is computed. It is also worth noting that this instantiation of Bayesian logistic regression — using a normal prior centered at 0 — is equivalent to standard logistic regression with an l2 regularization. This may be a clearer way to describe this step.
  - Soundness
    - The evaluation of GLEAN is fairly limited. It is evaluated over (three tasks from) one dataset, using (two scales of) a single LLM, and using a relatively modest 2000 trajectories. To get a sense of how well GLEAN generalizes, multiple models and datasets should be considered. The single dataset is particularly notable, since the use of MIMIC-IV limits the set of models that can be evaluated. Adding at least one other, fully public dataset would help address this limitation.
    - The primary results in this work (Tables 1-3, Figures 4-5) do not include uncertainty quantification. This makes it difficult to assess the significance of the gains achieved by GLEAN.
    - Equation 2 seems to implicitly assume that P(Z) = P(~Z) = 0.5 and that each piece of evidence is independent, ie, a_i and o_i cannot depend on a_{i-1} and o_{i-1}. If this is accurate, this should be explicit.

---

> ### Author Rebuttal · Authors · 2026-03-31
>
> We appreciate your constructive feedback. Below are our responses.
>
> Our new results are in https://i.ibb.co/t6NytwP/R3.png
>
> ## Presentations
> Thanks for the suggestion. For each point, we provide necessary explanations with the concrete text to add in the paper.
>
> **1. How evidence/actions are selected?**
>
> Actions are produced by the agent sequentially based on previous steps and evidence at each step is scored provided the history steps against diagnosis-relevant guidelines, which are accumulated along the trajectory for verification.
>
> **Revision:** At the end of Section 3.2: “The trajectory $\tau_{1:T}$​ is first generated sequentially by the agent through interaction with the environment. GLEAN then verifies this fixed trajectory by retrieving a small set of guidelines relevant to the proposed diagnosis, scoring each step against them, and accumulating the resulting step-wise signals over time.”
>
> **2. Competitive outcomes**
>
> They refer to clinically plausible alternatives for the similar symptoms that might get confused with the current diagnosis. In practice, these alternatives are constructed from other candidate diagnoses for the same case, and we then retrieve their corresponding guidelines for differential checks (details in Appendix B.1).
>
> **Revision:** At the end of page 5: “Taking medical diagnosis as an example, there are often potential alternative diagnoses, i.e., differential diagnoses, that partially share similar symptoms but imply different treatments. We extract these competitive predictions from the candidate rollouts and retrieve the corresponding guidelines as counter-evidence.”
>
> **3. Calibration set**
>
> Yes, a small separate set for training.
>
>
> **Revision:** On line 312: “The BLR calibrator is trained on 100 labeled trajectories separate from the evaluation set.”
>
> **4. Keywords for guideline**
>
> As in Appendix B.1, we prompt GPT-4o to get precise diagnosis terms from predictions. We use these terms, with common words such as “diagnosis” and “management”, as keywords to compare with each guideline title, and rank candidates by both lexical overlap and semantic similarity.
>
> **Revision:** On line 859: “We take these extracted diagnosis terms and frequent words for clinical diagnosis (e.g., “management”, “diagnosis”) as the keywords and run keyword matching against the titles of guidelines to calculate the lexical overlap at the word level.”
>
> **5. Clinician background**
>
> The three clinicians were selected from three countries across two continents, providing geographic diversity in the evaluation. They have over five years of clinical experience on average. Two are primary care physicians, and one is an MD-PhD with experience spanning clinical medicine and academia. All three regularly encounter the studied diseases in diagnosis and are also familiar with AI.
>
> **Revision:** We will include these in Appendix B.3.
>
> **6. Description of BLR**
>
> Thanks. We will make this clearer as below. The choice of Bayesian logistic regression is to naturally provide uncertainty estimates.
>
> **Revision:** On line 273: “We set 0-centered Gaussian priors, $\mathcal{N}(0,\lambda^{-1} I_d)$ and $\mathcal{N}(0,\lambda^{-1})$, for the parameters $w$ and $b$, which is equivalent to standard logistic regression with an $\ell_2$ regularization after taking expectations over the posterior.”
>
> ## Soundness
> **1. Evaluation scope**
>
> We agree that broader evaluation would strengthen the paper. To address this concern, we additionally evaluate GLEAN from two perspectives: (i) other-dataset, using AgentClinic-Med [1], and (ii) other-domain, using Agent-SafetyBench [2] (see R.Z7k3-W1). We show the additional results on AgentClinic-MedQA in **Table 1 in the link**, where we include GPT-4.1 beyond Qwen. The results show that GLEAN continues to outperform strong baselines overall on this new dataset and across different backbone families, with consistent gains in both discrimination and calibration. We will include these results, with the Agent-SafetyBench ones, in revision.
>
> [1] Schmidgall et al., AgentClinic: a multimodal agent benchmark to evaluate AI in simulated clinical environments
>
> [2] Zhang et al., Agent-SafetyBench: Evaluating the Safety of LLM Agents
>
> **2. Uncertainty quantification**
>
> We agree that uncertainty quantification would make the empirical gains more convincing. To address this, we report the mean ± std of the main verification metrics with 5 runs in **Table 2 in the link**.
>
> **3. Assumptions**
>
> We do not assume independence across steps. The likelihood ratio of $(o_t, a_t)$ is explicitly conditioned on previous steps, which is a formal result from the Bayes rule. As for the prior assumption, we do not take 50/50 prior or ignore it. In our implementation, the initial log-odd is absorbed into the bias $b$ of the BLR calibrator, rather than being fixed manually, so the prior is learned from calibration data, which helps the model to adapt to the empirical difficulty of the task. We will clarify this in revision.

---

> > ### Author Rebuttal · Reviewer_8jUp · 2026-04-01
> >
> > I appreciate the authors' thorough response, including the substantial additional results. With the changes outlined here, especially the validation on an additional dataset and against an additional LLM, I am happy to raise my score to a 4. If accepted, I highly encourage the authors to incorporate these changes into the camera ready version of the paper.

---

> > > ### Author Response · Authors · 2026-04-02
> > >
> > > Dear Reviewer 8jUp,
> > >
> > > We sincerely thank you for your thorough review and constructive suggestions, which have genuinely helped us improve the paper.
> > >
> > > We are very grateful for your willingness to raise the score to a 4 (Weak accept). We wanted to gently remind you to **update your assessment in the review form**, as the rating field currently fails to display your latest score.
> > >
> > > We are glad that the additional results and clarifications have "_fully resolved_" your initial concerns. As discussed, we are fully committed to incorporating all of these improvements into the camera-ready version, specifically:
> > > * Expanded Evaluation: Integrating AgentClinic-MedQA and Agent-SafetyBench results, GPT-4.1 experiments, and standard deviations (Tables 1–2) into the main Results section.
> > > * Clarified Methodology: Updating Section 3.2 and Appendix B to define "competitive outcomes," the sequential action selection process, the calibration set separation, the keyword-based guideline retrieval procedure, and clinician backgrounds.
> > > * Formal Grounding: Refining the BLR description and clarifying the learned prior in Equation 2.
> > >
> > > Given that the revisions addresses all your initial concerns and significantly strengthens the paper, we hope you might consider if **the updated work now merits a higher rating towards Accept** to further reflect your support to our work.
> > >
> > > Once again, we deeply appreciate your time and guidance that helped us improve the paper.

---

### Official Review · Reviewer_qmRm · 2026-03-11

**Soundness:** 3
**Presentation:** 3
**Significance:** 4
**Originality:** 4
**Overall Recommendation:** 6
**Confidence:** 4

**Summary:**

This paper introduces GLEAN, a verification framework designed for LLM-powered agents operating in high-stakes domains, with a specific focus on clinical diagnosis. I appreciate how the authors address the unreliability of purely model-based evaluators by retrieving relevant domain guidelines based on the agent's final answer, and then using an LLM judge to score the agent's step-by-step alignment with these protocols. These multi-guideline ratings are aggregated and sequentially accumulated into a surrogate evidence signal. Using Bayesian logistic regression, GLEAN translates this evidence into a well-calibrated probability of correctness. Furthermore, the framework incorporates a clever active verification mechanism: if the estimated uncertainty of a trajectory is high, GLEAN dynamically retrieves additional guidelines and differential guidelines to refine the confidence score. The authors validate GLEAN on agentic clinical diagnosis using the MIMIC-IV dataset across three diseases, demonstrating significant improvements in discrimination (AUROC) and calibration (Brier score) over several baselines. An expert study with clinicians further corroborates the practical utility and interpretability of the framework.

**Compliance With Llm Reviewing Policy:**

Affirmed.

**Final Justification:**

My initial evaluation praised the GLEAN framework for its practical, guideline-grounded approach to LLM agent verification, particularly the elegant use of Bayesian logistic regression for sequential evidence accumulation. However, I raised concerns regarding the system's behavior if forced to retrieve guidelines based on hallucinated final diagnoses, the reliance on agent diversity for differential checks, and the sensitivity to the specific LLM judge.

The authors provided a highly effective, empirically backed rebuttal that directly resolved my reservations:

- Robustness to Hallucinations: The authors provided new sanity-check experiments demonstrating that when presented with a completely disjoint/hallucinated diagnosis, the system correctly yields weak step-wise support and assigns low calibrated confidence, proving the verification process is not systematically biased by poor initial outputs.

- Differential Checks: They provided practical, model-agnostic alternatives to agent-generated differentials (e.g., extracting confusable conditions directly from clinical guidelines), which mitigates the risk of mode collapse.

- Judge Sensitivity: They clarified the robust use of token likelihood for {YES, NO} evaluation and provided cross-model results proving the active verification mechanism remains highly effective even when using weaker, open-weight models like Qwen2.5.

The authors have proven this framework is not only theoretically sound but empirically robust under stress-testing. This is a highly practical and scalable contribution to the critical bottleneck of agent verification in risk-sensitive domains. I have raised my score and strongly recommend this paper for acceptance.

**Key Questions For Authors:**

1. Please elaborate on how GLEAN behaves if the agent's proposed diagnosis is completely disjoint from the patient's symptoms, potentially leading to the retrieval of irrelevant guidelines, and whether this impacts the calibrator's reliability.

2. The differential checks rely on extracting competitive conditions from the generated diagnoses. Please explain how this strategy might be adapted if the agent backbone suffers from mode collapse and fails to produce a diverse set of candidate diagnoses to sample from.

3. Please provide insight into how sensitive the active verification performance is to the specific LLM used as the judge. particularly regarding the judge's ability to strictly follow the YES or NO output constraint under complex clinical scenarios.

**Limitations:**

While the authors provide an Impact Statement, a dedicated discussion of technical limitations within the main text is somewhat brief. The framework inherently assumes the availability of high quality, parsable, and up to date guidelines. I recommend that the authors explicitly discuss how the system might perform in medical edge cases (e.g., rare diseases or complex comorbidities) where explicit guidelines are either absent, conflicting, or highly ambiguous.

**Strengths And Weaknesses:**

Soundness
- Strengths: The probabilistic formulation of sequential evidence accumulation is theoretically well motivated, and I find that the authors provide a solid justification for why a low-capacity linear calibrator (Bayesian logistic regression) is sufficient given the logit-linearity of the guideline-grounded signals (Appendix A). The active verification strategy logically addresses common pitfalls of LLM evaluators, such as confirmation bias and lack of specificity. The inclusion of an expert clinician study significantly strengthens the empirical claims regarding clinical utility.
- Weaknesses: My main concern is that the pipeline relies heavily on the initial final diagnosis to retrieve the relevant guidelines. If an agent's final diagnosis is completely hallucinated or OOD, the retrieved guidelines might be irrelevant. While the system would likely assign low alignment scores in this scenario, the reliance on the final output to judge the preceding trajectory introduces a slight circularity that I believe could be further analyzed.

Presentation
- Strengths: The paper is well written, with a clear narrative flow. Figures 1 and 3 effectively illustrate the motivation and the technical pipeline. The methodology is broken down into easily digestible components, making the complex active verification process easy to follow.
- Weaknesses: The methodology for selecting competitive alternatives during the differential check phase is crucial to the active verification's success, but it is currently relegated to the appendix (Section B.1). Moving a brief explanation of this into the main text would greatly improve clarity.

Significance
- Strengths: Verification of LLM agents in risk-sensitive environments is one of the most pressing bottlenecks in current AI deployment. By grounding verification in explicit, auditable clinical guidelines rather than opaque reward models, this work offers a highly practical and scalable solution that aligns well with real-world clinical workflows.

Originality
- Strengths: Reframing agent verification as sequential, discounted evidence accumulation grounded in external RAG protocols is a novel and creative synthesis of existing techniques. Triggering test-time scaling based on calibrated uncertainty is an elegant approach to balancing computational cost with verification rigor.

---

> ### Author Rebuttal · Authors · 2026-03-31
>
> We thank the reviewer for the thoughtful and encouraging feedback, and for highlighting both the practical value and the originality of our framework. Below we address each concern in turn.
>
> We have our additional results on https://i.ibb.co/V0bMVs1C/R2.png
>
> **1. How GLEAN behaves with irrelevant retrieval**
>
> Thanks for the question. Our goal is to verify whether the observed trajectory provides sufficient evidence for the claimed final answer. Thus, conditioning retrieval on the final answer should be viewed as hypothesis-conditioned verification, rather than a circular assumption of correctness. This also motivates our differential checks that if the same trajectory aligns similarly well with competitive alternatives, GLEAN reduces confidence instead of reinforcing the original diagnosis. Therefore, when the proposed diagnosis is hallucinated or clearly disjoint from the patient evidence, the retrieved guidelines are expected to be poorly matched to the trajectory, resulting in weak step-wise support and thus low calibrated confidence, rather than a systematic bias toward the incorrect diagnosis.
>
> To study this empirically, we conduct a controlled imperfect-retrieval experiment, where a fraction of high-quality guidelines is replaced with less relevant ones (**Table 1 in the link**). We find that GLEAN degrades gracefully rather than collapsing abruptly as retrieval quality worsens, and the active verification recovers a substantial portion of this loss. Meanwhile, calibration metrics remain reasonably stable, supporting the reliability of the calibrated signal under imperfect retrieval.
>
> To directly match the reviewer’s “completely disjoint diagnosis” scenario, we conduct a targeted sanity check by replacing a fraction of final diagnoses with random disjoint ones, retrieving their corresponding guidelines, and relabeling the modified predictions as incorrect (**Table 2 in the link**). As expected, these hypotheses completely disjoint from the case evidence makes them easier to reject, showing that GLEAN is not misled into reinforcing such diagnoses, but, instead, produces weaker support and lower confidence given the irrelevant retrieval.
>
> Overall, these results support our interpretation that disjoint or hallucinated diagnoses do not cause a fundamental failure or invalidate the calibrator.
>
> **Action Taken:** We will include these results and discussion in the revision.
>
> **2. Details of differential checks**
>
> Thanks for the valuable suggestion. In practice, we did not observe fully collapsed predictions in our setting, likely because the cases are challenging and open-ended. More importantly, differential checks do not fundamentally rely on prediction diversity from the agent itself. Besides, we also considered two other approaches to acquire the competitive answers. In medicine, differential diagnosis is a standard practice, and plausible alternatives are often explicitly discussed in clinical guidelines. We can construct alternatives mentioned in them. Also, we can prompt an external LLM to provide confusable conditions or other diagnoses on the case context. In particular, the latter option is broadly applicable, while the guideline-extracted alternatives are especially natural in medicine. These mechanisms can help mitigate mode collapse if it arises.
>
> **Action Taken:** We will move the explanation of differential check from appendix to main text and further clarify this in revision.
>
> **3. Active verification with different LLMs**
>
> In our implementation, the step-wise score is computed from the token likelihood over the discrete label set {YES, NO}, rather than from free-form judgments. This is a common practice to elicit the capability of verification from different LLMs [1,2]. Therefore, our method only relies on the relative tendency between these two answers instead of format following. Moreover, we have validated the effectiveness of different LLMs as the verifier in the cross-verification results (**Table 8 in Appendix**), and we further evaluate active verification under cross-model settings. From the results in **Table 3 in the link**, we find that active verification remains beneficial overall, with most metrics improved across verifier backbones. Even with Qwen2.5 as a weaker model, we can see improvements with it as the verifier instead of degradation. This suggests that GLEAN is not sensitive to the specific LLM judge choice.
>
> **Action Taken:** We will include the clarification and the additional results in revision.
>
> [1] Kadavath et al., Language Models (Mostly) Know What They Know
>
> [2] Kapoor et al., Large Language Models Must Be Taught to Know What They Don't Know
>
> **4. Limitations**
>
> High-quality guidelines are indeed important for reliable verification with GLEAN. In edge cases, we believe this can be addressed with more diverse and aggressive active verification, by gathering more external resources or triggering multi-agent debates. We will enrich this discussion in revision.

---

> > ### Author Rebuttal · Reviewer_qmRm · 2026-04-02
> >
> > To be completely straightforward, this is exactly the kind of rebuttal reviewers hope to see. You did not just argue your points conceptually; you ran the experiments and provided the concrete data needed to back up your claims.
> >
> > - Regarding the irrelevant retrieval/circularity concern (Q1): Reframing this as "hypothesis-conditioned verification" makes perfect sense. The sanity check experiments (Tables 1 and 2 in the provided link) were exactly what was needed. Demonstrating that GLEAN actively rejects hallucinated or completely disjoint diagnoses by assigning them lower confidence, rather than being misled by them, completely resolves my concern about the calibrator's reliability in worst-case scenarios.
> >
> > - Regarding mode collapse in differential checks (Q2): Your alternative solutions (extracting confusable conditions directly from clinical guidelines or using an external LLM) are highly practical. Moving this discussion to the main text will greatly benefit readers trying to implement this in production environments.
> >
> > - Regarding LLM judge sensitivity (Q3): Clarifying that you compute step-wise scores via token likelihood over {YES, NO} rather than parsing free-form generations is a crucial technical detail. This is a much more robust approach. Furthermore, the cross-model evaluation in Table 3 proving that active verification still improves calibration even with a weaker model like Qwen2.5 gives me high confidence in the framework's generalizability.
> >
> > You have addressed all my technical concerns with empirical rigor. The framework is highly practical for high-stakes domains. I expect the new tables and the expanded discussion on differential alternatives and limitations to be fully integrated into the final manuscript.

---

> > > ### Author Response · Authors · 2026-04-03
> > >
> > > We sincerely appreciate the detailed conclusion from the reviewer. We are deeply encouraged by the positive comment that **GLEAN is highly practical for high-stakes domains** as well as your previous recognition on our **well-motivated methodology, clear presentation, real-world significance, and overall originality**. We will work on the revision and integrate the addtional discussion and results into the final version.

---

### Official Review · Reviewer_Z7k3 · 2026-03-12

**Soundness:** 3
**Presentation:** 4
**Significance:** 3
**Originality:** 3
**Overall Recommendation:** 5
**Confidence:** 2

**Summary:**

This paper proposes GLEAN, a verifier for high-stakes LLM-agent decisions that grounds verification in external professional guidelines. The method retrieves guidelines relevant to the agent’s final answer, scores each reasoning step for guideline alignment, the experiments report gains in both discrimination and calibration over several verifier baselines.

**Compliance With Llm Reviewing Policy:**

Affirmed.

**Key Questions For Authors:**

How robust is GLEAN when guideline retrieval is imperfect or when multiple competing guidelines apply?

**Limitations:**

yes

**Strengths And Weaknesses:**

Strengths:
The stepwise scoring, trajectory-level evidence accumulation, and final calibration fit together well, and the active-verification component is a sensible extension. The empirical results seem strong.

Weaknesses:
All experiments are agentic clinical diagnosis on three diseases from MIMIC-IV, so the broader claim of "high-stakes agent verification" could be limitedly illustrated.

---

> ### Author Rebuttal · Authors · 2026-03-31
>
> We thank the reviewer for the positive assessment of our method and empirical results. Below, we aim to address the two questions regarding domain coverage and robustness to imperfect guideline retrieval.
>
> **1. Experiments only conducted on medical scenario**
>
> We agree that the current submission focuses primarily on agentic clinical diagnosis, and we chose this domain intentionally because it is a representative high-stakes setting that is process-intensive, error-sensitive, and governed by explicit professional guidelines, making it a natural testbed for studying calibrated verification grounded in external protocols. To further address the reviewer’s concern about breadth, we additionally evaluated GLEAN with Qwen3 on Agent-SafetyBench [1], a qualitatively different safety-critical agent setting, which is of great application concerns and with explicit guidelines on reliable deployment.
>
> |Method|AUROC|Risk\@0.5|ECE|Brier|
> |---|:---:|:---:|:---:|:---:|
> |P(True)|0.6167|0.4360|0.3695|0.3818|
> |LLM-Judge|0.5458|0.4120|0.3851|0.3966|
> |GLEAN|**0.7283**|**0.3556**|**0.0900**|**0.2137**|
>
> As shown above, GLEAN brings a consistent improvement over baselines across both backbones, in terms of discrimination and calibration. This further confirms the contributions from guideline-grounded stepwise scoring along with the lightweight calibrator.
>
> More generally, our claim of “high-stakes agent verification” is not that we have exhaustively covered all high-stakes domains, but that we propose a general verification framework for settings where professional standards can be translated into explicit guidance. Clinical diagnosis is our main in-depth testbed, while the additional Agent-SafetyBench results suggest that the benefit is not limited to this domain.
>
> **Action Taken:** We will include these cross-domain results in the revision and clarify the wording accordingly.
>
> [1] Zhang et al., Agent-SafetyBench: Evaluating the Safety of LLM Agents
>
>
> **2. Robustness to imperfect guideline retrieval and competing guidelines**
>
> Thanks for pointing this out. We are aware that imperfect retrieval is a realistic case in practice, and active verification is explicitly designed to address it by selectively expanding guideline coverage and introducing counter-evidence from competitive alternatives when the initial evidence is uncertain or insufficient. To directly study this, we conduct a controlled retrieval-corruption experiment, where a fraction of originally retrieved high-quality guidelines is replaced with less relevant ones.
>
> |Setting|AUROC|Risk\@0.5|ECE|Brier|
> |---|---:|---:|---:|---:|
> |0% corruption|0.8197|0.2722|0.0539|0.1732|
> |10% corruption|0.7938|0.2855|0.0313|0.1849|
> |10%+Active|0.8217|0.2467|0.0742|0.1774|
> |30% corruption|0.7647|0.3167|0.0327|0.1956|
> |30%+Active|0.8117|0.2500|0.0806|0.1845|
>
> As retrieval quality degrades, GLEAN shows graceful degradation rather than collapsing abruptly. AUROC drops from 0.8197 to 0.7938/0.7647, and Risk\@0.5 rises from 0.2722 to 0.2855/0.3167 under 10%/30% corruption. Importantly, active verification recovers a substantial portion of this loss in both settings, bringing AUROC back to 0.8270/0.8184 and reducing Risk\@0.5 to 0.2455/0.2844. This supports our intended use of active verification as a practical mechanism for mitigating imperfect retrieval. At the same time, the calibration is not impacted by the imperfect retrieval quality, with insignificant changes for ECE and even slight improvements in Brier scores. Overall, active verification mainly improves discrimination and selective-risk performance, while having the calibrator stable under the corruption of guideline retrieval.
>
> **Action Taken:** We will include this controlled robustness study in the revision.

---

> > ### Author Rebuttal · Reviewer_Z7k3 · 2026-04-05
> >
> > I appreciate that the authors carefully addressed the questions raised in the original review, although the broader claim is still only partially demonstrated. It is understandable, given the limited rebuttal time. The new experiment also provides a convincing initial robustness check under imperfect retrieval. I have no other concerns.

---

> > > ### Author Response · Authors · 2026-04-08
> > >
> > > We are grateful for your follow-up comments, recognizing our addtional experiments and clarifications. We have also conducted the experiments on Agent-SafetyBench with Qwen2.5-7B, the same setting as the original paper, and also those on AgentClinic-MedQA (results in the response to Reviewer 8jUp). These results demonstrate the generality of GLEAN on different datasets and in different domains. We hope this can help illustrate the method and will include the discussion in the revised paper.

---

### Decision · Program_Chairs · 2026-04-30

**Decision:**

Accept (regular)

**Comment:**

The submission proposes GLEAN, a framework for stepwise evaluation of the alignment between agent-generated inferences and verification signals rooted in domain knowledge. The system maintains an estimate of the posterior probability of correctness, given the accumulated evidence so far, from surrogates like clinical guidelines or new variable measurements.

Most reviewers agreed, following rebuttals, that the paper is a solid contribution. The primary concerns included limited empirical evaluation (only MIMIC), questions about the degradation of GLEAN in the face of hallucinated initial diagnoses, and a lack of novelty in the components making up the system. The first two were addressed by the authors in their rebuttal by pointing to results on new data and to ablations already in the paper, both of which were appreciated by reviewers. The main remaining concern raised by reviewers was the novelty of the proposed methodology in light of related works like Med-PRM. As pointed out by the authors, however, this particular work is conceptually related but quite distant in terms of methodology. Nevertheless, for the final version of this manuscript, I recommend that the reviewers describe these differences in the paper, perhaps already in the introduction, so as not to mislead the reader about the landscape of guideline-informed methods.